# AutoScale: Automatic Prediction of Compute-optimal Data Composition for Training LLMs

## Abstract

Domain reweighting is an emerging research area aimed at adjusting the relative weights of different data sources to improve the effectiveness and efficiency of language model pre-training. This paper demonstrates that the optimal composition of training data from different domains is scale-dependent, challenging the existing practice of determining optimal mixtures through small-scale experiments and directly applying them at larger scales. We derive an analytical model for the dependence of optimal weights on data scale and introduce AutoScale, a novel, practical approach for optimizing data compositions at potentially large training data scales. AutoScale first uses a principled optimization framework to find optimal compositions at smaller, feasible scales, then predicts optimal compositions at larger scales using our derived model. Our evaluation on GPT-2 Large and BERT pre-training demonstrates AutoScale's effectiveness in improving training convergence and downstream performance. Particularly, for GPT-2 Large on RedPajama, AutoScale decreases validation perplexity 28% faster than baselines, with up to 38% speed-up over unweighted training, achieving the best performance across downstream tasks. This work provides insights into the varying benefits of data sources across training scales for language models, contributing to the burgeoning research on scale-dependent data curation. Code is open-sourced[1].

## 1 Introduction

Large language models (LLMs) are pre-trained on vast datasets sourced from diverse domains. However, the immense computational demands of this process, coupled with limited resources, create a pressing need to enhance the effectiveness and efficiency of pre-training. A promising approach to address this challenge is through *domain reweighting*—adjusting the relative proportions of data from different sources (1–6).

Showing encouraging potentials, though, current implementation techniques face significant limitations. A prevailing technique is to first optimize data composition for a smaller proxy model and at a smaller data scale (1; 5–7). Yet, this optimization often employs alternative objectives that may not align with the primary goal of minimizing evaluation loss. Moreover, the optimized weights are directly applied to training the target model on much larger data scales, implicitly assuming that the "optimal data composition" remains constant across data scales. This assumption of scale-invariance, however, may not hold in practice, potentially leading to suboptimal performance when scaling up. While research has scale-dependent data selection at the individual data point level for vision models (8; 9), it remains unclear whether this scale dependence applies to domain-level optimization, or how such scaling behavior might manifest in language models.

In parallel, an increasingly popular practice is to directly adopt domain weights that were designed for training previous models (10), such as those used for LLaMA (11). However, these weights are optimized for specific applications that may differ from the desired use case of the target model. Given the limitations of these approaches, many in the industry still rely on heuristics for mixing domain data (12; 10; 13). These limitations highlight the ongoing need for more adaptive and scale-aware methods to determine effective domain weights across various model sizes and target scenarios.

---

[1] https://anonymous.4open.science/r/AS25

This work explicitly investigates and confirms the scale-dependence of optimal domain mixing, characterizing its scaling behavior. Based on these findings, we develop a practical methodology: optimizing domain weights at smaller, affordable scales and leveraging derived scaling laws to predict optimal mixing at much larger target scales. We lay out an overview of this work and main results in Fig. 1. Our contributions are summarized as follows.

① **Principled algorithmic framework for optimal domain mixing.** Investigating the scaling law of domain mixing extends beyond mere empirical study. It requires a mathematical definition of optimal domain mixing and a tractable algorithm to solve for optimal weights. Our first contribution is formulating the optimal mixing problem as a bi-level optimization. However, existing general bi-level optimization techniques (14; 15) are intractable in this context due to their reliance on second-order information. We propose a novel approach tailored to our problem context that leverages scaling laws to estimate the dependence of the learned model's loss on the weights, effectively reducing the bi-level problem to a single level. Our algorithm requires retraining models only linearly in the number of data domains, making it feasible for exploratory studies.

② **Uncovering and quantifying the scale-dependence of optimal domain composition.** Leveraging the algorithm developed in ①, we conduct empirical studies to optimize domain weights at different training data scales. Our results demonstrate that the optimal data composition varies with the scale of the training data, suggesting that the common practice of empirically determining an optimal composition using small-scale experiments will not yield optimal data mixtures for larger scales. We further derive an analytical framework for modeling the functional relationship between optimal data composition and training data scales.

③ **Practical algorithm for optimal domain mixing.** While the algorithm in ① has made optimal domain mixing feasible for exploratory studies, its retraining requirements limit its practicality to smaller scales. To enable data composition optimization at large scales, we propose AUTOSCALE. This method works by finding optimal data compositions at smaller, computationally feasible scales, fitting a predictor using our analytical model for the scale-dependency of optimal composition mentioned in ②, and finally using this predictor to determine optimal data composition at larger scales. Since one only needs to train models on small data scales where re-training is affordable, AUTOSCALE does not require using proxy models with a smaller parameter size, avoiding transferability issues between domain weights optimized with different model sizes.

④ **Robust performance gains across models and datasets.** Our evaluation of AUTOSCALE on both decoder-only and encoder-only models demonstrates its consistent ability to achieve significant computational savings. For instance, in pre-training `GPT-2 Large` (16) on the `RedPajama` dataset, AUTOSCALE decreases validation perplexity 28% faster than any baseline, with up to 38% speed-up compared to training without reweighting. It also achieves the best overall performance across downstream tasks. Additionally, we present intriguing findings regarding the varying benefits of traditionally perceived high-quality and low-quality data sources across different training scales. Specifically, we observe that data sources with standardized formats, such as `Wikipedia` and scientific papers—often regarded as high-quality—are most beneficial at smaller scales but exhibit sharp diminishing returns as the training data scales up. Conversely, with increased compute, data sources containing diverse examples, such as `CommonCrawl`, demonstrate continued reductions in training loss even at considerably large training data scales.

## 2 RELATED WORK

**Domain Reweighting.** An emerging line of research strives to optimize the composition of training data for LLMs pre-training with *domain reweighting* , i.e., adjusting the relative proportion of data from different data sources to "best" (in terms of training efficiency, final model performance, etc.) train the model. DOREMI (1) first trains a small reference model, and then trains a second proxy model with GroupDRO (17) to minimize the excessive domain loss relative to the reference model, where the domain weights of the proxy model will be the output. DOGE (5) trains a proxy model while tracking the first-order gradient of the model on evaluation domains (i.e., data influence) and optimizes domain weights based on the gradients, relying on infinitesimal approximations which may or may not be accurate for models trained with a practical learning rate. DATA MIXING LAWS (6) trains a number of proxy models to run a coarse grid search on the space of data mixtures and interpolate their performance with exponential functions to find the minimum. Similarly, RegMix(7)

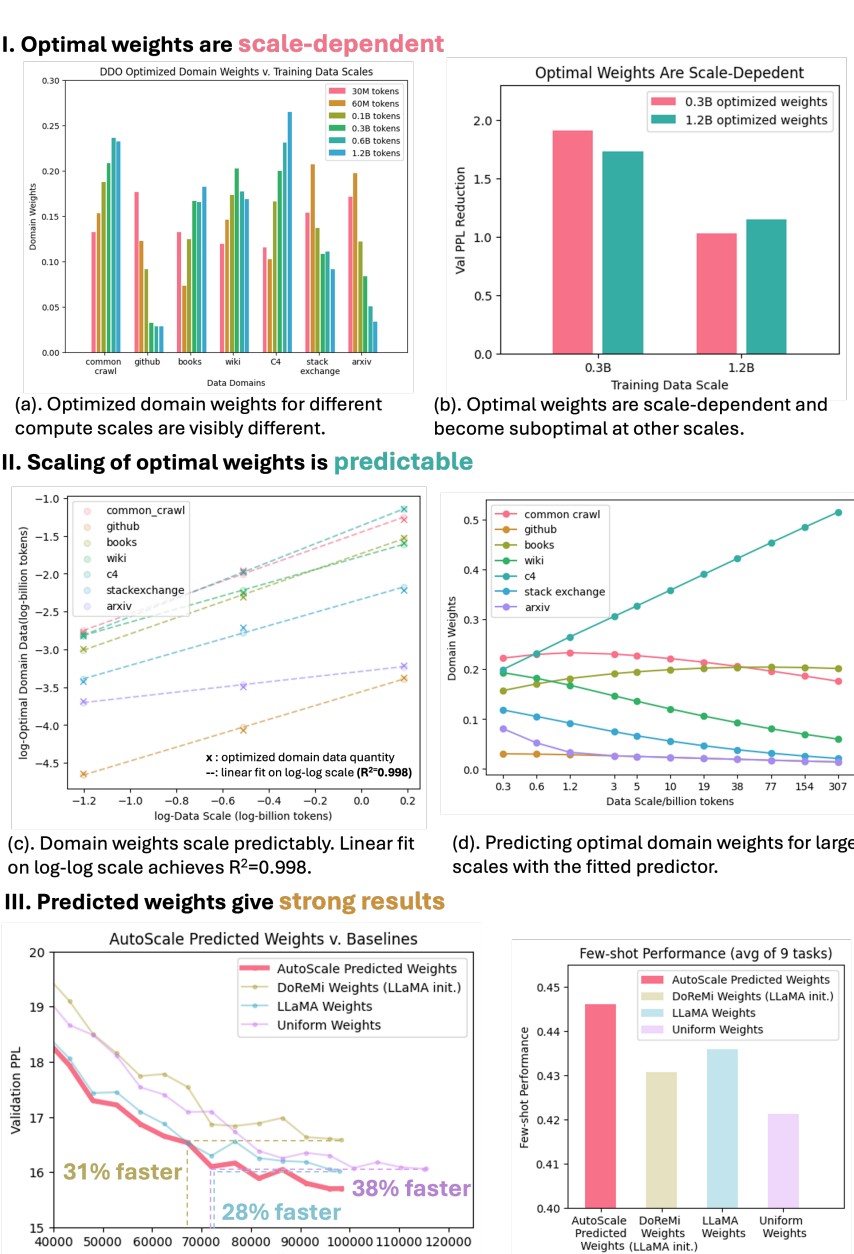

Figure 1: Overview and main results. I. Optimizing domain weights with the proposed Direct Data Optimization (DDO) algorithm for pre-training 774M Decoder-only LMs (`GPT-2 Large`). Optimal domain weights depend on the scale of training data. A consistent shift can be observed (*data sources with standardized formats, such as* `Wikipedia` *and scientific papers—often regarded as high-quality—are most beneficial at smaller scales but exhibit sharp diminishing returns as the training data scales up. Conversely, with increased compute, data sources containing diverse examples, such as* `CommonCrawl`, *demonstrate continued reductions in training loss even at considerably large training data scales.*). Using domain weights optimized for a different scale yields sub-optimal results, failing to fully realize the benefits of domain reweighting. II. Optimal domain data quantity (y-axis) for different training data scales (x-axis) shows high linearity ($R^2 = 0.998$) on log-log plot, suggesting the shifting pattern can be well predicted by exponential-style functions. We fit AUTOSCALE to predict optimal domain weights for larger training scales. As we scale up, data sources with diverse samples (e.g., `C4`) are upweighted relative to domains with standard format (e.g., `Wikipedia`). III. Training 774M Decoder-only LMs for 10B tokens (96k steps). AUTOSCALE-predicted domain weights decrease validation PPL at least $28\%$ faster than any baseline with up to $38\%$ speed up, achieving best overall task performance.

trains a regression model to represent the relationship between training data mixtures and resulting model performance and optimize data composition based on it.

These methods often rely on ad-hoc hyperparameter tuning via trial and error, achieving varying results. *Further, the optimized weights are directly applied to training the target model on magnitudes of larger data scales.* This implicitly poses a strong assumption that the "optimal data composition" is invariant of model sizes or data scales. Yet, optimal data composition is likely to shift with data size. *Optimal curation at a smaller scale may not remain optimal at the target scale* (8; 9). (18) provides a recent survey for this fast-evolving field. We refer to App. B for broader discussions.

**Scaling Laws.** Extensive research shows that *Neural Scaling Laws*, predicting how the model performance changes with the scale of training data, model parameters, and computation budget (19), to be accurate in various tasks from vision and text processing (20) to LLM pre-training (13) and evaluations (21). (22) proposes compute-optimal scaling for LLM pretraining data scales together with the model's parameter sizes. Yet, recent progress (23; 10) shows no sign of saturation in pre-training even for models pre-trained on a considerably larger data scale than recommended by (22). (24) shows that data from different sources generally scale at different rates. Seminal work (8) sheds light on the possibility of attaining beyond-neural scaling law performance if one could find the best training dataset for each training data scale. This work is connected to the research on scaling laws in two ways. First, we leverage scaling laws to model the functional relationship between the quantity of data from each domain and trained model performance, allowing optimizing the training data composition in a reasonable time with high precision; further, *this work contributes to a novel dimension of scaling laws–scaling optimal data compositions with the training data scale, providing original insights, clear empirical evidence, and theoretical frameworks which enable further analysis.*

# 3 OPTIMAL DATA COMPOSITION IS SCALE-DEPENDENT AND PREDICTABLE

For "compute-optimal" domain weights, the goal is to find an optimal training data composition such that, for a given compute budget (i.e., training data size), the empirical validation loss, measured in perplexity (PPL), is minimized (1; 3; 5). Formulating this as a bi-level optimization problem, in this section, we first introduce an original solution approach via scaling law approximations, which allows solving it efficiently and effectively. Then, with this solution approach, we solve for the optimal domain weights under different training data scales. Our results demonstrate that the optimal data composition for a fixed compute budget depends on the scale of the training data. Via the lens of scaling laws, this work pioneers in deriving an analytical framework for modeling the functional relationship between optimal data composition and training data scales.

## 3.1 COMPUTE-OPTIMAL TRAINING DATA COMPOSITIONS

Consider training an LLM on a data composition $S$ from $m$ domains, $D_1, D_2, \cdots, D_m$. Let $S = \{S_1, S_2, \cdots, S_m\}$ denote the training dataset where $S_i$ is the subset of training data from each domain. The domain weights $\mathbf{w} = [w_1, w_2, \cdots, w_m]^T$ are defined as the proportions of data for each domain. Namely, letting $N = |S|$ denote the amount of total tokens of training data, domain weights are given as $w_i = N_i/N$, where $N_i$ denotes the amount of tokens for training subset $S_i$.

Let $\boldsymbol{\theta}^*(S)$ denote the parameters of a learning algorithm (i.e., the model) trained on data $S$ with empirical risk minimization (ERM), given as $\boldsymbol{\theta}^*(S) := \arg\min_{\boldsymbol{\theta}} \mathcal{L}(\boldsymbol{\theta}, S)$ where $\mathcal{L}(\boldsymbol{\theta}, S)$ denotes the loss of model parameterized by $\boldsymbol{\theta}$ evaluated on data $S$, which is the training loss. Since training data $S$ can be equivalently defined by its data quantity and domain weights $(N, \mathbf{w})$, we define a slight change of notation $\boldsymbol{\theta}^*(N, \mathbf{w}) := \boldsymbol{\theta}^*(S)$ and will use $S$ and $(N, \mathbf{w})$ interchangeably. We would like to maximize the amount of information gain and achieve maximum loss reduction during training, given as $\min_{\mathbf{w} \in \mathbb{W}^m} \mathcal{L}(\boldsymbol{\theta}^*(N, \mathbf{w}), D^v) = \min_{\mathbf{w} \in \mathbb{W}^m} \sum_{i=1}^{m} \mathcal{L}(\boldsymbol{\theta}^*(N, \mathbf{w}), D_i^v)$, where $D^v$ and $D_i^v$ denote total validation data and validation data of individual domain $i$, respectively; the space of weights $\mathbb{W}^m$ is the hyperplane of the probability simplex $\mathbb{W}^m = \{\mathbf{w}|w_1 + w_2 + \cdots + w_m = 1\} \cap \{\mathbf{w}|0 \leq w_i \leq 1, \forall i \in \{1, 2, \cdots, m\}\}$. Define minor simplifications of notations for the validation losses $\mathcal{L}^v(\theta, D^v) := \mathcal{L}(\theta, D^v)$ and $\mathcal{L}_i^v(\theta, D^v) := \mathcal{L}(\theta, D_i^v)$. Then, the optimal domain weights, $\mathbf{w}^*$, are given as the minimizer of the objective,

$$\mathbf{w}^* = \arg\min_{\mathbf{w} \in \mathbb{W}^m} \sum_{i=1}^{m} \mathcal{L}_i^v(\boldsymbol{\theta}^*(N, \mathbf{w})) \quad \text{s.t.} \quad \boldsymbol{\theta}^*(N, \mathbf{w}) = \arg\min_{\boldsymbol{\theta}} \mathcal{L}(\boldsymbol{\theta}, (N, \mathbf{w})) \tag{1}$$

where perplexity is adopted as the loss metric. This formulation is a bi-level optimization problem, where the outer problem seeks the optimal domain weights, while the inner problem is training the

model with ERM on the data defined by certain weights. A general approach is to solve it with gradient descent, $\mathbf{w}^{t+1} = \mathbf{w}^t - \eta \cdot \frac{\partial \mathcal{L}_v(\boldsymbol{\theta}^*(N, \mathbf{w}^t))}{\partial \mathbf{w}}$. Since there is no tractable form of analytical expression for $\boldsymbol{\theta}^*$, this gradient needs to be estimated with empirical methods (e.g., approximated by finite difference), requiring repetitive re-training of the model at each update (25).

### 3.1.1 SOLUTION VIA SCALING-LAW-INSPIRED APPROXIMATIONS

Directly optimizing training data composition by solving bi-level optimization problems involves repetitive model retraining, which can be prohibitively expensive even at small scales. Current work (1; 5–7) mostly employs heuristic methods to conduct this optimization on smaller models trained with fewer data, achieving varying results in different cases. To crystalize the relationship between optimal data compositions and training data scales and obtain a clear image of the complete landscape, we propose an *original* approach to this problem. We propose to first fit a scaling function to the outer loss (validation loss) $\mathcal{L}^v$ as a function of domain weights $\mathbf{w}$, effectively reducing the bi-level problem to a single level, which can be solved efficiently via regular gradient descent, allowing finding the global optimum efficiently and accurately.

To begin with, neural scaling laws suggest the relationship between a model's evaluation loss and the size of its training data can be well-represented by power law functions (19) $\mathcal{L}_v(\boldsymbol{\theta}^*(N, \mathbf{w})) = N^{-\gamma} + \ell_0$ where constants $\ell_0$ denotes some irreducible loss and $\gamma \geq 0$ is some scaling coefficient. Drawing inspirations from (26), which formulates the scaling laws for transfer learning, we propose the following approximation to model the scaling relationship between model loss and training data quantity from different sources/domains.

Consider a model trained on data with size $N$ and domain weights $\mathbf{w}$. Define constant $N_0^i$ which estimates the evaluation loss when the amount of training data from domain $i$ is zero (i.e., $N_i' = 0$), which effectively measures the effect of data from all other domains. From this regard, $N_0^i$ can be interpreted as the *equivalent data size* for training data from domains other than $i$. Notably, this formulation aligns with empirical findings in the prior literature (26; 6). Then, for training data defined by $(N, \mathbf{w})$ where the amount of training data from *domain $D_i$* is $N_i = N \cdot w_i$, evaluation loss can be expressed as a function of $N_i$: $\mathcal{L}_v(\boldsymbol{\theta}^*(N, \mathbf{w})) = (N_0^i + N_i)^{-\gamma_i} + \ell_i$ where $\gamma_i, \ell_i$ are constants associated with domain $i$. If the amount of training data from *one domain $D_i$* is changed from $N_i$ to $N_i'$ with the amount of training from other domains unchanged, we approximate the new model's evaluation loss, $\mathcal{L}_i'$, after re-training with a power law function of $N_i'$:

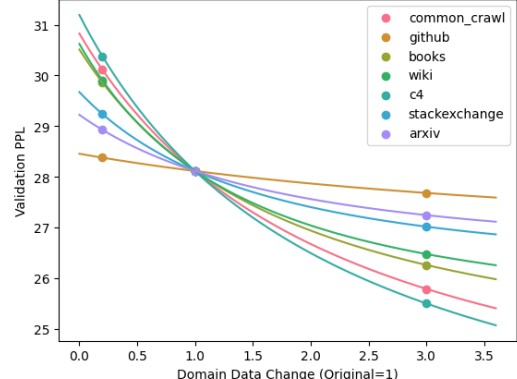

Figure 2: Fitting validation loss with power-law functions for 774M Decoder-only LMs (`GPT-2 Large`), directly approximating how loss changes with each domain's data quantity. (*X-axis depicts the quantity of domain data relative to the original amount before perturbation (e.g., 1.0=100%).*)

$$\mathcal{L}_v(\boldsymbol{\theta}^*(N', \mathbf{w}')) = (N_0^i + N_i')^{-\gamma_i} + \ell_i := \mathcal{L}_i' \tag{2}$$

where $N' = N + (N_i' - N_i)$ denotes the updated amount of training data, and $w_i' = N_i'/N'$ denotes the updated domain weights.

We propose the following procedure to fit the parameters in Eq. (2). We re-train two models with different data quantities for domain $i$, $N_i^+$ and $N_i^-$ where $N_i^- < N_i < N_i^+$, and compute their evaluation loss, $\mathcal{L}_i^+$ and $\mathcal{L}_i^-$, respectively[2]. Then, together with evaluation loss $\mathcal{L}_v^0 = \mathcal{L}_v(\boldsymbol{\theta}^*(N, \mathbf{w}))$ for the original model trained with $N_i$, the parameters $\gamma_i, \ell_i$ and $N_0^i$ can be estimated via ordinary least square (OLS) fitting,

$$N_0^i, \gamma_i, \ell_i = \arg \min_{N_0^i, \gamma_i, \ell_i} [\mathcal{L}_v^0 - (N_0^i + N_i)^{-\gamma_i} - \ell_i]^2 + [\mathcal{L}_i^+ - (N_0^i + N_i^+)^{-\gamma_i} - \ell_i]^2 + [\mathcal{L}_i^- - (N_0^i + N_i^-)^{-\gamma_i} - \ell_i]^2 \tag{3}$$

Compared to the original model, the difference in evaluation loss *due to the change of data for domain $D_i$* is given as $\mathcal{L}_v(\boldsymbol{\theta}^*(N', \mathbf{w}')) - \mathcal{L}_v(\boldsymbol{\theta}^*(N, \mathbf{w})) = (N_0^i + N_i')^{-\gamma_i} - (N_0^i + N_i)^{-\gamma_i}$. Repeating this

---

[2]Empirically, we found setting the perturbation ratio, $r = N_i/N_i^- = N_i^+/N_i = 3$, produces reliable results.

process and fitting the scaling functions *for each domain*, finally, we express the evaluation loss as a function of the amount of data from each domain as their summation: $\mathcal{L}_v(\boldsymbol{\theta}^*(N', \mathbf{w}')) - \mathcal{L}_v(\boldsymbol{\theta}^*(N, \mathbf{w})) = \sum_{i=1}^{m} \left[ (N_0^i + N_i')^{-\gamma_i} - (N_0^i + N_i)^{-\gamma_i} \right]$ where $N' = N + \sum_i (N_i' - N_i)$ and $w_i' = N_i'/N'$. *Empirically, evaluation loss is shown to be well represented by such function form as depicted in Fig. (2)*, which shows fitting validation loss with the proposed power-law functions for training 774M Decoder-only LMs (`GPT-2 Large`), directly approximating how loss changes with each domain's data quantity. Results with Encoder-only LMs (`BERT`) demonstrates the same trend (Fig. 10(b)). This representation lends us an analytical form for the desired objective.

To derive the final objective, we add the constraint for the total amount of training data to be the same as before, i.e., $\sum_i N_i' = N' = N$, which explicates our interest in reweighting data from each domain without changing the training data size. Then, by definition, domain data quantity $N_i' = w_i' \cdot N' = w_i' \cdot N$. Note that $(N_0^i + N_i)^{-\gamma_i}$ is independent of $\mathbf{w}'$, making it orthogonal to the optimization problem. Finally, our problem becomes

$$\mathbf{w}^* = \arg\min_{\mathbf{w}' \in \mathbb{W}^m} \sum_{i=1}^{m} \left[ (N_0^i + N_i')^{-\gamma_i} - (N_0^i + N_i)^{-\gamma_i} \right] = \arg\min_{\mathbf{w}' \in \mathbb{W}^m} \sum_{i=1}^{m} (N_0^i + w_i' \cdot N)^{-\gamma_i}.$$

Since the objective is defined as the summation of convex functions, we end up with a convex optimization problem. With the constraint on the probability simplex and the objective being easily differentiable, the problem can be solved extremely efficiently using *projected gradient descent* (27). We term this solution approach as DDO Algorithm (Direct Data Optimization). We provide its pseudocode below and an operational pipeline in App. C.

---

**Algorithm 1** Direct Data Optimization (DDO)

---

**Require:** $m$ domains (data sources) with data $D_1 \dots D_m$, data budget $N_0$ ($\ll$ for full-scale training), training dataset $S$, model parameters $\boldsymbol{\theta}$, validation loss $\mathcal{L}_v$, perturbation ratio $r > 1$ (e.g., $r = 3$).
 Initialize weights for all domains $\forall i \in \{1, \dots m\}$: $w_i \leftarrow 1/m$;
 Initialize training data for all domains $\forall i \in \{1, \dots m\}$: sample $S_i \subset D_i$ where $|S_i| = w_i \cdot N$;
 Train the model on data $S = \{S_1 \dots S_m\}$ and evaluate its loss $\mathcal{L}_v^0 \leftarrow \mathcal{L}_v(\boldsymbol{\theta}^*(S))$;
 **for** $j$ from 1 to $m$ **do**
  $w_j^+ \leftarrow r \cdot w_j$;           $\triangleright$ Perturb domain weights (+)
  Resample $S_j^+ \subset D_j$ where $|S_j^+| = w_j^+ \cdot N$;
  Train the model on data $S = (\{S_1 \dots S_m\} \setminus S_j) \cup S_j^+$ and evaluate its loss $\mathcal{L}_j^+ \leftarrow \mathcal{L}_v(\boldsymbol{\theta}^*(S))$;
  $w_j^- \leftarrow \frac{1}{r} \cdot w_j$;          $\triangleright$ Perturb domain weights (-)
  Resample $S_j^- \subset D_j$ where $|S_j^-| = w_j^- \cdot N$;
  Train the model on data $S = (\{S_1 \dots S_m\} \setminus S_j) \cup S_j^-$ and evaluate its loss $\mathcal{L}_j^- \leftarrow \mathcal{L}_v(\boldsymbol{\theta}^*(S))$;
  OLS fit for scaling functions $N_0^i, \gamma_i, \ell_i = \arg\min_{N_0^i, \gamma_i, \ell_i} [\mathcal{L}_v^0 - (N_0^i + N_i)^{-\gamma_i} - \ell_i]^2 + [\mathcal{L}_i^+ - (N_0^i + N_i^+)^{-\gamma_i} - \ell_i]^2 + [\mathcal{L}_i^- - (N_0^i + N_i^-)^{-\gamma_i} - \ell_i]^2$;
 **end for**
 Output optimized domain weights $\mathbf{w}^* = \arg\min_{\mathbf{w}' \in \mathbb{W}^m} \sum_{i=1}^{m} (N_0^i + w_i' \cdot N)^{-\gamma_i}$.

---

## 3.2 Optimal Data Compositions are Scale-Dependent

With the DDO algorithm, for a fixed model training pipeline and data sources, we conducted empirical studies to optimize domain weights at different training data scales. Our results demonstrate that the optimal data composition for a fixed compute budget depends on the scale of the training data, suggesting that the common practice of empirically determining an optimal composition using small-scale experiments will not yield optimal data mixtures for larger models.

Fig. 1(a)(b) shows the results on optimizing domain weights with DDO algorithm for pre-training 774M Decoder-only LMs (`GPT-2 Large`). Optimal domain weights depend on the scale of training data. A consistent shift can be observed. Using domain weights optimized for a different scale yields sub-optimal results, failing to realize the benefits of domain reweighting fully. *These results clearly show that the hypothesis, "optimal data composition is invariant of data scales", implicitly assumed by many existing works, is largely untrue.* On the contrary, a consistent pattern can be observed in how optimal data compositions shift with the scale of training data. For example, data sources with standard format such as `Wikipedia` and scientific papers, regarded as high quality, are most beneficial at smaller scales but observe sharp diminishing returns as training data scales up. With more compute, data sources with diverse examples, such as `CommonCrawl`, demonstrate continued reductions in training loss even for larger training data scales.

Foreshadowed by (8; 9), beyond-neural scaling law performance might be attained if one could find the best training dataset for each training data scale. This treasure chest remains unexplored in the

context of training LLMs. *This consistent pattern of shifts suggests predictability in the relationship between optimal composition and training data scales*, holding the promise to unlock substantial improvements in training efficiency and model performance.

### 3.3 DERIVING SCALING LAWS FOR PREDICTING OPTIMAL DATA COMPOSITION

Following the above findings, this work pioneers in deriving an analytical framework for modeling the functional relationship between optimal data composition and training data scales. Via the lens of scaling laws, the analysis lays out theoretical foundations that could be of independent interest.

Recall that neural scaling laws give the relationship between evaluation loss and training data quantity as $\mathcal{L} = N^{-\gamma} + \ell_0$ where $\mathcal{L}$ is the evaluation loss (e.g., perplexity), $\ell_0$ denotes some irreducible loss, and $\gamma \geq 0$ are some constant. $(\ell_0, \gamma)$ can be fitted empirically. Without loss of generality, consider a standard case where the evaluation metric is aggregated loss over multiple independent tasks where each training sample will only contribute to a single task and the loss of each task only scales with the amount of training data contributing to this task as a power law function. Then, for a total of $m$ tasks, the aggregated evaluation loss scales as the following $\mathcal{L} = \ell_0 + \sum_{i=1}^{m} \beta_i \cdot N_i^{-\gamma_i}$, where $\ell_0$ denotes some irreducible loss, $N_i$ denotes the quantity of data contributing to task $i$ and constants $\beta_i \geq 0$ and $\gamma_i \geq 0$ are coefficients associated with task $i$. Define diagonal matrix $\mathbf{N} = diag\{N_1, N_2, \cdots N_m\}$. For a training data scale $N = \sum_i N_i$, define compute-optimal data composition $\mathbf{N} = diag\{N_1^*, N_2^*, \cdots N_m^*\}$ as the minimizer of $\mathcal{L}$, given as $\mathbf{N}^* = \arg\min_{\sum_i N_i = N} \ell_0 + \sum_{i=1}^{m} \beta_i \cdot N_i^{-\gamma_i}$. We propose the following theorem, which states the optimal data composition scales in exponential-style functions with the amount of training data and can be directly predictable from that of smaller scales.

**Theorem 1.** *Consider the following optimization problem* $\min_{\mathbf{N}} \left\{ \sum_{i=1}^{m} \beta_i N_i^{-\gamma_i} \middle| \sum_{i=1}^{m} N_i = N \right\}$.

*For any two compute budgets $N^{(1)} \neq N^{(2)}$, let $\mathbf{N}^{(1)*}$ and $\mathbf{N}^{(2)*}$ be their respective minimizers. Then, for any third compute budget $N^{(3)}$ such that $N^{(1)} \neq N^{(3)} \neq N^{(2)}$, the corresponding minimizer $\mathbf{N}^{(3)*}$ must satisfy $\mathbf{N}^{(3)*} = \mathbf{N}^{(2)*}(\mathbf{N}^{(1)*})^{-1}\mathbf{N}^{(2)*}$.*

See App. D.1 for the formal theorem statement and a complete proof. Examples for illustration are also provided in D.1. We built our theory from a standardized example which assumes the evaluation metric is composed of independent tasks with separate scaling laws. In App. D.2, we further extend this theory to a general case where the same conclusion can be reached without the independence assumption, where we consider the evaluation to be composed of a number of *independent* sub-tasks ("latent skills" (28)) which are hidden variables. Finally, we note that empirical results are shown to be highly consistent with the derivations above: In Fig. 1(c), optimal domain data quantity (y-axis) for different training data scales (x-axis) shows high linearity ($R^2 = 0.998$) on log-log plot, *suggesting the shifting pattern can be well-described by the exponential-style function forms described above.*

## 4 TOWARDS A PRACTICAL ALGORITHM FOR FINDING OPTIMAL COMPOSITIONS

In this section, we introduce a practical paradigm for finding optimal data compositions developed based on the theoretical analyses and empirical insights introduced above. Having shown the consistent pattern of shifts in optimal data composition with the scale of training data and unveiled its predictability from scaling law analysis, moving forward, this paper presents a novel tool–AUTOSCALE, which automatically predicts optimal training data compositions at larger scales based on compositions optimized at smaller scales.

---

**Algorithm 2** AUTOSCALE

**Require:** Optimal domain weights (obtained from DDO) $\mathbf{w}^{(1)*}$ at data scale $N^{(1)}$ and $\mathbf{w}^{(2)*}$ at data scale $N^{(2)}$, target data scale $N^{(t)}$, where $N^{(1)} < N^{(2)} < N^{(t)}$.
  Optimal domain data $\mathbf{N}^{(1)*} \leftarrow \mathbf{w}^{(1)*} \cdot N^{(1)}$;
  Optimal domain data $\mathbf{N}^{(2)*} \leftarrow \mathbf{w}^{(2)*} \cdot N^{(2)}$;
  Next optimal domain data $\mathbf{N}^{(\mathbf{x})*} \leftarrow \mathbf{N}^{(2)*}(\mathbf{N}^{(1)*})^{-1}\mathbf{N}^{(2)*}$;
  Next data scale $N^{(x)} \leftarrow \sum_i N_i^{(x)*}$;
  **while** $N^{(x)} < N^{(t)}$ **do**
    Next optimal domain data $\mathbf{N}^{(\mathbf{x})*} \leftarrow \mathbf{N}^{(2)*}(\mathbf{N}^{(1)*})^{-1}\mathbf{N}^{(2)*}$;
    Next data scale $N^{(x)} \leftarrow \sum_i N_i^{(x)*}$;
  **end while**
  Output predicted optimal domain weights $\mathbf{w}^{(\mathbf{t})*} \leftarrow \mathbf{N}^{(\mathbf{x})*}/N^{(x)}$.

---

Theoretical analysis above shows that the optimal quantity for each domain scales in *exponential-style functions* with training data size. We leverage this result to enable the automatic prediction of optimal training data compositions at larger scales from optimal compositions at small scales. First, for

two smaller training data scales $N^{(1)}$ and $N^{(2)}$ where $N^{(1)} \neq N^{(2)}$, find their optimal training data compositions $\mathbf{N}^{(1)*}$ and $\mathbf{N}^{(2)*}$ where $\sum_i N_i^{(1)*} = N^{(1)}$ and $\sum_i N_i^{(2)*} = N^{(2)}$ using DDO algorithm provided in Sec. 3. Models trained at scales $N$ and $N'$ are considered proxy models where re-training these models is affordable. Since $N^{(1)}$ and $N^{(2)}$ are small data scales where re-training these models is affordable, AUTOSCALE *does not require using proxy models with a smaller parameter size, avoiding the transferability issue between domain weights optimized on different models.* Then, $\mathbf{N}^{(1)*}$ and $\mathbf{N}^{(2)*}$ yield the optimal training data composition at the next scale as $\mathbf{N}^{(3)*} = \mathbf{N}^{(2)*}(\mathbf{N}^{(1)*})^{-1}\mathbf{N}^{(2)*}$, where $N_i^{(3)*} = (N_i^{(2)*})^2/N_i^{(1)*}$ is the optimal amount of training data for each domain. This gives that for data scale $N^{(3)} = \sum_i N_i^{(3)*}$, optimal domain weights are given as $w_i^{(3)*} = N_i^{(3)*}/N^{(3)}$. Then, $\mathbf{N}^{(3)*}$ can be combined with either $\mathbf{N}^{(1)*}$ or $\mathbf{N}^{(2)*}$ to make the next prediction. Repeat this process until the target training data scale is reached. The procedure is described in the pseudocode above with an operational pipeline provided in App. C.

## 5 EMPIRICAL RESULTS

Two sets of empirical studies are conducted: Causal Language Modeling (CLM) in Sec. 5.2, and Masked Language Modeling (MLM) in Sec. 5.3. We train models with up to 10B tokens and report the number of steps saved to reach the same evaluation loss (perplexity). We also report downstream task performance to benchmark performance improvements after training the same number of steps.

### 5.1 EXPERIMENTAL SETUP

In Sec. 5.2, we pretrain 774M Decoder-only LMs (`GPT-2 Large` architecture (16)) **from scratch** on the `RedPajama` dataset (29). `RedPajama` dataset is an open-source reproduction of the training data used for LLaMA-1/2 models (11), totaling 1.2T tokens from 7 data domains with proportions: `Common Crawl` (67%), `C4` (30) (15%), `GitHub` (4.5%), `Wikipedia` (4.5%), `ArXiv` (2.5%), and `StackExchange` (2.0%). In Sec. 5.3, we pretrain 110M Encoder-only LMs (`BERT-base` architecture (31)) **from scratch** on data from 5 typical sources—`Amazon Reviews`, `Arxiv`, `Books`, `Wikipedia`, and `Open WebText Corpus` (32). Further details are in App. E.1 and E.2. Runtime and GPU hours are documented in App. E.7.

### 5.2 CAUSAL LANGUAGE MODELING WITH DECODER-ONLY LMS (GPT)

**Evaluation** We test the perplexity on the held-out dataset, comprising 10K samples each from the 7 domains. For downstream tasks, we include: `BoolQ` (33) (zero-shot), `HellaSwag` (34) (zero-shot, 10-shot), `PIQA` (35) (zero-shot), `TruthfulQA` (36) (zero-shot), `PubMedQA` (37) (10-shot), `CrowsPairs` (38) (25-shot), and `ARC-Easy` (39) (zero-shot). Additionally, `BBH Novel Concepts` (40) task is added to the aggregated results for models trained beyond 10B tokens, making a total of 9 tasks. We select tasks that ensure the model's performance surpasses random guessing, spanning from question answering and commonsense inference to bias identification and scientific problem solving. These tasks provide a comprehensive assessment of model performance (10; 21). We adopt the evaluation framework from (41). More details are available in App. E.4.

**Baselines** We report results for our methods (DDO and AUTOSCALE) and 5 baselines–UNIFORM, LLAMA WEIGHTS (curated), DOREMI (LLaMA weights initialization), DATA MIXING LAWS from (6) and DOREMI from (1) (uniform initialization). Uniform weights uniformly sample data from all domains, resulting in the same number of training tokens from each domain. LLaMA weights are a set of curated domain weights heuristically tuned for training LLaMA-1/2 models. We implemented DOREMI proposed in (1). DOREMI trains two smaller-scale auxiliary models (proxy models). First, a reference model is trained with the dataset's original domain weights, which are the LLaMA weights for `RedPajama` dataset. Then, optimized domain weights are obtained by using a proxy model to minimize the worst-case excess loss across different domains. We train both auxiliary models for 50K steps. Implementation details are available in App. E.3. Besides, we compare with 2 domain weights from existing literature, which are optimized on the same dataset, `RedPajama`, with similar Decoder-only LMs. DATA MIXING LAWS (6) first performs a grid search on the space of possible data mixtures and records evaluation loss for proxy models trained on these mixtures. Then, the loss is interpolated with exponential functions to find the optimal domain weights for the proxy model. DOGE (5) also implements DOREMI (1) with auxiliary models trained for 50K steps but with the reference model trained with uniform weights. We evaluate the model trained on these domain weights to present a complete landscape.

**Direct Data Optimization (DDO):** We conduct DDO Algorithm to optimize domain weights for models (774M Decoder-only LMs) trained from scratch with 30M to 1.2B tokens. *As depicted in Fig. 1(a), optimal domain weights for each training data scale are visibly different and demonstrate a clear shifting pattern.* We found data sources with standard format such as `Wikipedia` and scientific papers, regarded as high quality, are most beneficial at smaller scales and observe sharp diminishing returns as the training data scales up. With more compute, data sources with diverse examples, such as `CommonCrawl`, continue to reduce training loss for even larger training data scales. In Fig. 1(b), we validated this observation in Fig. 1(b), where we trained two models with 0.3B tokens with domain weights optimized at 0.3B tokens and 1.2B tokens, and two models with 1.2B tokens with these weights, respectively. *Takeaway 1: the results show that, the optimal weights are only optimal at the scale it is optimized and become suboptimal when applied on other scales.*

**Predicting Optimal Weights at Larger Scales with AUTOSCALE:** With DDO-optimized weights from models trained up to 0.6B tokens, we fit AUTOSCALE predictor and use it to visualize how the optimal domain weights will shift as we continue scaling up training data. Depicted in Fig. 1(d) and Fig. 6, as the training data scale grows, data sources with diverse examples, such as `C4` and `CommonCrawl`, will take up a considerable proportion of training data. Therefore, we expect LLaMA weights will perform better when the training data scale is sufficiently large. The results also suggest training on data from `Books` domain will continue to provide benefits. *Takeaway 2: thus, AUTOSCALE-predicted domain weights give a larger weight to Books domain compared to baselines which counterintuitively downweight high-quality book contents.*

| Method/Task | truthfulqa_mc2 | pubmedqa | piqa | hellaswag (10-shot) | crows_pairs_english | boolq | arc_easy | hellaswag (zero-shot) | Avg |
|---|---|---|---|---|---|---|---|---|---|
| Uniform Weights | 0.4526 | 0.438 | 0.6115 | 0.2923 | 0.5886 | 0.5636 | 0.3742 | 0.2907 | 0.4514 |
| LLaMA Weights | 0.434 | 0.492 | 0.6055 | 0.2944 | **0.5903** | 0.5612 | 0.3956 | 0.2952 | 0.4585 |
| Data Mixing Laws (ref) | **0.4537** | 0.468 | 0.6061 | 0.2951 | 0.5778 | **0.6162** | 0.3771 | 0.2938 | 0.4610 |
| DoReMi (ref) | 0.4505 | 0.468 | 0.5985 | 0.2886 | 0.5742 | 0.5410 | 0.3750 | 0.2896 | 0.4482 |
| **AutoScale (ours)** | 0.4385 | **0.536** | **0.6202** | **0.3021** | 0.585 | 0.6141 | **0.3977** | **0.303** | **0.4746** |

Table 1: Downstream task performance for 774M Decoder-only LMs trained for 3B tokens. Models trained with AUTOSCALE-predicted weights achieve the best overall performance across the tasks.

Subsequently, to examine AUTOSCALE-predicted weights, we train models on larger scales with 3B, 5B, and 10B tokens. On 3B training data, we compare AUTOSCALE-predicted weights with Uniform weights, LLaMA weights, DOREMI weights from (5) (uniform initialization), and DATA MIXING LAWS weights from (6). In both 3B and 5B results (Fig. 7), AUTOSCALE *achieves the lowest validation perplexity after the same steps, at least* 25% *faster than any baseline with up to* 37% *speed up.* Provided in Table 6, AUTOSCALE-predicted weights significantly reduced the loss on `Books` domain and also achieved much lower worst-domain perplexity. When testing the few-shot performance on 8 downstream tasks, the model trained with AUTOSCALE-predicted weights achieves the best overall performance (Table 1). Results for models trained with 10B tokens are depicted in Fig. 1(e)(f), where we added the comparison with DOREMI initialized with LLaMA weights. *Takeaway 3:* AUTOSCALE-*predicted weights consistently outperform any baseline with a* 28% *to* 38% *margin and demonstrate advantageous performance on downstream tasks.* Echoing our predictions, as training data scales up, LLaMA weights visibly outperform uniform domain weights. See App. E.5 for additional results .

### 5.3 MASKED LANGUAGE MODELING WITH ENCODER-ONLY LMS (BERT)

We evaluate the model's MLM loss on held-out validation datasets, comprising 10K samples each from the 5 training domains. Additionally, as an auxiliary evaluation, we test the MLM loss on 3 non-training held-out domains. To be consistent with the perplexity loss used in CLM, we report the exponential cross-entropy loss for MLM. We evaluate the model's task performance on `GLUE` benchmark (42) (with 8 diverse tasks for natural language understanding (NLU)) and `SQuAD` (43) (a large-scale QA dataset). See App. E.4 for more details. Uniform weights are used as the baseline.

**Direct Data Optimization (DDO):** We conduct DDO Algorithm to optimize domain weights for proxy models (110M Encoder-only LMs) trained from scratch with MLM on 1GB data. Results for DDO-optimized weights are shown in Fig. 3. *Takeaway 3a: DDO visibly decreased the model's validation loss on all training domains as well as held-out non-training domains, demonstrating its effectiveness in improving training efficiency and model utility.* When testing on `GLUE` *benchmark*

*and* `SQuAD` *dataset, consistent with the reduced evaluation loss,* DDO-optimized weights are shown to improve the model's performance on downstream tasks by a notable margin.

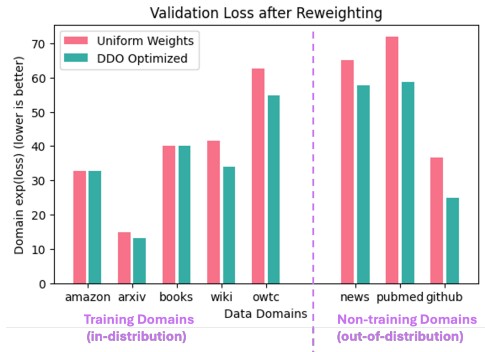 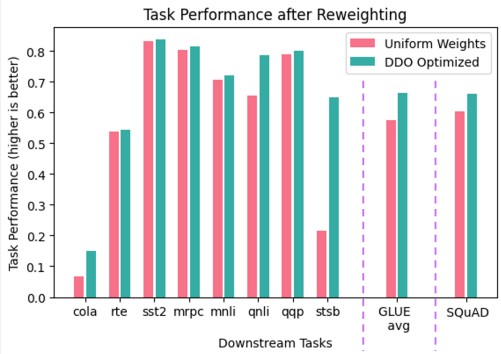

(a) Validation Loss (↓ lower is better)  (b) Task Performance (↑ higher is better)

Figure 3: Optimizing domain weights with DDO algorithm for pre-training Encoder-only LMs (`BERT`). DDO substantially reduces validation loss. After reweighting, all training domains' loss either decreased or remained unchanged. Out-of-domain loss on non-training domains also decreased considerably. Enhanced performance is observed on all `GLUE` tasks (eval metric: `cola`: Matt. corr., `stsb`: Pearson corr., rest: acc.) and `SQuAD` (acc.).

**Predicting Optimal Weights at Larger Scales with AUTOSCALE:** With DDO-optimized weights from models trained up to 0.5B tokens, we fit AUTOSCALE predictor and use it to predict how the optimal domain weights will shift as we continue scaling up training data. Depicted in Fig. 11, similar to the pattern described above, as the training data scale grows, data sources with diverse examples, such as WebText and Amazon Reviews, become increasingly important over standard domains, such as Wikipedia and Arxiv. One hypothesis is such data sources contain samples on diverse topics and language styles, providing rich information compared to domains with clean, standard text. We train models with MLM for up to 288k steps ($\sim 120\%$ of the pertaining data size for original `BERT-base` (44)). Table 7 shows that, compared to without reweighting (uniform weights), AUTOSCALE-*predicted weights speed up training by 16.7% on most data scales with a 10% speedup on the largest scale, validating its consistent effectiveness. Takeaway 4: nonetheless, the speedup is less impressive than in the results for Decoder-only LMs, demonstrating the different response to domain reweighting for models with different architecture or language modeling objectives.* This is also hinted in Fig. 10(b), where the evaluation loss has a similar response to data from different domains, suggesting limited potential for performance improvements from domain reweighting.

## 6 CONCLUSIONS

In this work, we demonstrate that the optimal data composition for a fixed compute budget varies depending on the scale of the training data, showcasing that the common practice of empirically determining an optimal composition using small-scale experiments will not yield the optimal data mixtures when scaling up to the final model. Addressing this challenge, we propose AUTOSCALE, an automated tool that finds a compute-optimal data composition for training at any desired target scale. In empirical studies with pre-training 774M Decoder-only and Encoder-only LMs, AUTOSCALE decreases validation perplexity at least 28% faster than any baseline with up to 38% speed up compared to without reweighting, achieving the best overall performance across downstream tasks.

**Limitations** & **Future Work**  The promising results achieved by AUTOSCALE in optimizing data composition for large-scale language model pretraining open up some intriguing avenues for future exploration. (1) *Generalizability*: It will be interesting to extend this research to larger-scale settings, other data modalities, and more comprehensive evaluation benchmarks, and re-examine the validity of insights provided by experiments at the scale that we work on. (2) *Direct optimization of downstream performance*: In practice, the capabilities of LLMs are characterized by their performance on various downstream tasks, and the perplexity loss that we focused on in this study is only a rough, inaccurate proxy for downstream performance. It will be interesting to extend AUTOSCALE to directly optimize downstream performance. (3) *More fine-grained data curation*: AUTOSCALE works with fixed data domains and only optimizes how the domains are mixed together, confining the optimization space. Intuitively, if one can strategically select the corpus within each domain and even adapt the data selection strategy to different stages of training, further improvements could be achieved.

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

# Appendices

## APPENDIX A  BROADER IMPACTS

Reducing the complexity and resource requirements associated with pretraining LLMs, AUTOSCALE contributes to the democratization of AI. Smaller organizations, academic institutions, and individual researchers can more easily participate in cutting-edge AI research and development, fostering innovation and collaboration across the AI community. Moreover, learning from massive amounts of data requires large and costly computational resources, which not only consume substantial energy but also generate a significant carbon footprint, contributing to environmental issues. Furthermore, these resources quickly become obsolete due to the rapid pace of technological advancements, leading to e-waste. This research makes contributions to mitigating these issues by improving the efficiency of resource utilization in AI training.

## APPENDIX B  EXTENDED RELATED WORK

**Training Data Curation** Data selection problems have been extensively studied for a variety of applications such as vision (45–48), speech (49; 50), and language models (45; 48; 51), and have been attracting growing interest over recent years. For LLMs, a line of research focuses on *data selection for pre-training* (also known as *pre-training data curation*) (52; 53; 22) from scratch or *continued pre-training*. (53) shows that continuing pre-training the model on the domain-specific dataset improves its performance on tasks of this domain; (54) uses importance resampling on simple bi-gram features with 10k bins to select millions of samples for domain/task adaptive pre-training. Problem-specific heuristic methods employ simple criteria to distinguish data quality for a given language model on particular datasets (e.g., via perceived relevance between how the dataset is created and training objectives of the LLM (55)). The effectiveness of these methods for data selection is often limited to specific use cases and easily fails when migrated to different problems (54). More recently, (56) selects samples for fine-tuning pre-trained LLMs via gradients of Optimal Transport distance. (57) curates pre-training data using GPT-4 to rate and select samples based on a number of quality criteria; further, (58) uses pre-trained LLMs to re-write the entire training corpus to improves its quality for pre-training other LLMs. (18) provides a recent survey for this fast-evolving field. Pre-training data curation is also studied for multimodal foundation models (MMFM)–e.g., (12) for vision-language models (VLMs), and (59; 9) for CLIP (Contrastive Language-Image Pretraining). Aside from pre-training LLMs, domain reweighting problems have been studied in research on collecting data for vision, audio, and text applications (60–63).

Besides, **Coresets** (64; 65) aim to find a representative subset of samples to speed up the training process, which may be formulated as an optimization problem. This process is considerably computationally intensive and hard to be applied on a practical scale for language applications. **Data Valuation** methods aim to measure the contribution of each sample to the model performance, which naturally provides a viable tool for data selection. Notable examples includes model-based approaches Shapley (66; 67), LOO (67; 68), and model-agnostic methods (69; 70). Achieving fruitful results in their respective applications and providing valuable insights, though, these methods are commonly known for their scalability issues. Model-based approaches require repetitive model training and often struggle to apply to a few thousand samples. A recent example, (71) uses a sampling approach to speed up a Shapley-style method for selecting data for fine-tuning LLMs and scales up to selecting from 7.28k subsets. It is hardly imaginable to apply it to the scale of practical language datasets. (69) utilizes the gradients of an Optimal Transport problem to provide an efficient measure of data values, yet the selection based on gradients does not necessarily align with the target distribution, resulting in mediocre performance in general cases.

## APPENDIX C  OPERATIONAL PIPELINE FOR ALGORITHMS

**Operational Pipeline (DDO)**

1. Train a base proxy model with uniform weights (or reference weights, if available);

2. At each time, add/reduce data quantity for one domain and re-train the proxy model;

3. Fit power law scaling functions and solve the optimization problem;

4. Iterate the process if necessary.

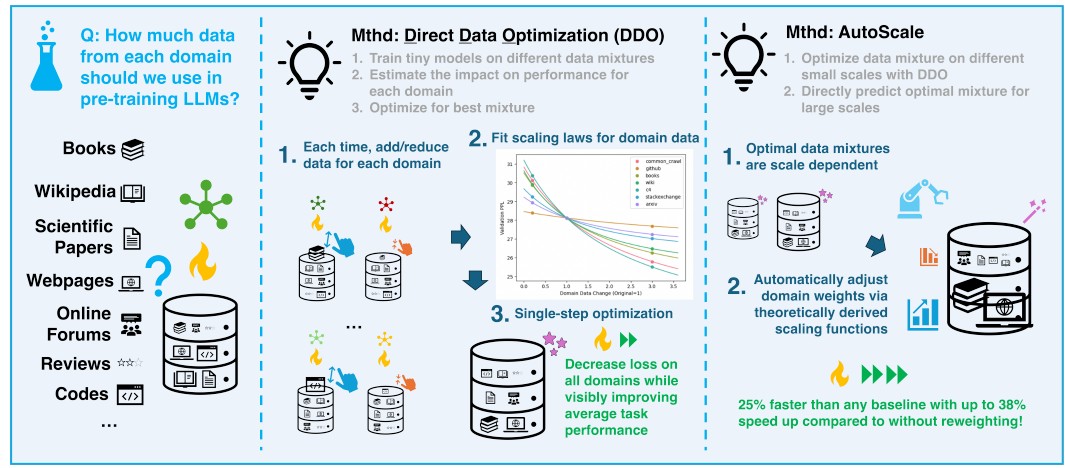

Figure 4: LLMs are pre-trained using data from different sources or domains, yet determining the optimal data composition is challenging. We propose AUTOSCALE, an automated tool that finds a compute-optimal data composition for training at any desired target scale. AUTOSCALE first determines the optimal composition at a small scale using a novel bi-level optimization framework, Direct Data Optimization (DDO), and then fits a predictor to estimate the optimal composition at larger scales. The predictor's design is inspired by our theoretical analysis of scaling laws related to data composition, which could be of independent interest. In empirical studies, AUTOSCALE decreases validation perplexity at least 25% faster than any baseline with up to 38% speed up compared to without reweighting, achieving the best overall performance across downstream tasks.

**Operational Pipeline (AUTOSCALE)**

1. For two smaller training data scales $N^{(1)}$ and $N^{(2)}$ where re-training the model is affordable, find their corresponding optimal training data compositions $\mathbf{N}^{(1)*}$ and $\mathbf{N}^{(2)*}$ using DDO Algorithm described above;

2. Predict the next optimal training data composition as $\mathbf{N}^{(3)*} = \mathbf{N}^{(2)*}(\mathbf{N}^{(1)*})^{-1}\mathbf{N}^{(2)*}$, yielding optimal domain weights $w_i^{(3)*} = N_i^{(3)*}/N^{(3)}$ at new training data scale $N^{(3)} = \sum_i N_i^{(3)*}$;

3. Repeat this process until the target training data scale is reached.

## APPENDIX D  PROOFS FOR SEC. 4

### D.1  THEOREM 1: SCALING LAW FOR OPTIMAL DATA COMPOSITIONS

**Theorem 1** (Scaling Law for Optimal Data Compositions **(restated)**). *Consider the following optimization problem*

$$\min_{\mathbf{N}} \left\{ \sum_{i=1}^{m} \beta_i N_i^{-\gamma_i} \,\middle|\, \sum_{i=1}^{m} N_i = N \right\}$$

*For any two compute budgets $N^{(1)} \neq N^{(2)}$, let $\mathbf{N}^{(1)*}$ and $\mathbf{N}^{(2)*}$ be their respective minimizers. Then, for any third compute budget $N^{(3)}$ such that $N^{(1)} \neq N^{(3)} \neq N^{(2)}$, the corresponding minimizer $\mathbf{N}^{(3)*}$ must satisfy $\mathbf{N}^{(3)*} = \mathbf{N}^{(2)*}(\mathbf{N}^{(1)*})^{-1}\mathbf{N}^{(2)*}$.*

*Proof.* For the evaluation loss given as

$$\mathcal{L} = \sum_{i=1}^{m} \beta_i \cdot N_i^{-\gamma_i},$$

at an optimal data composition, KKT condition (27) gives that we have the partial derivative of the loss function w.r.t. the amount of data from each domain equal to each other. This gives, for any two domains $a$ and $b$ (w.l.o.g, we simplify the derivation to the case of two domains) with optimal data quantity $N_a^*$ and $N_b^*$, respectively, we have

$$\frac{\partial \mathcal{L}}{\partial N_a} = -\beta_a \cdot \gamma_a \cdot N_a^{-\gamma_a - 1}$$

$$\frac{\partial \mathcal{L}}{\partial N_b} = -\beta_b \cdot \gamma_b \cdot N_b^{-\gamma_b - 1}$$

$$\left. \frac{\partial \mathcal{L}}{\partial N_a} \right|_{N_a = N_a^*} = \left. \frac{\partial \mathcal{L}}{\partial N_b} \right|_{N_b = N_b^*}$$

With straightforward derivations, this gives

$$-\beta_a \cdot \gamma_a \cdot (N_a^*)^{-\gamma_a - 1} = -\beta_b \cdot \gamma_b \cdot (N_b^*)^{-\gamma_b - 1}$$

$$\frac{\beta_a \cdot \gamma_a}{\beta_b \cdot \gamma_b} = \frac{(N_a^*)^{\gamma_a + 1}}{(N_b^*)^{\gamma_b + 1}}$$

$$(N_a^*)^{\gamma_a + 1} = \frac{\beta_a \gamma_a}{\beta_b \gamma_b} (N_b^*)^{\gamma_b + 1} \tag{4}$$

$$N_a^* = \left[ \frac{\beta_a \gamma_a}{\beta_b \gamma_b} (N_b^*)^{\gamma_b + 1} \right]^{\frac{1}{\gamma_a + 1}}$$

Let $N_a^{(2)*}, N_b^{(2)*}$ be the optimal data quantity for domains $a$ and $b$ at a *different data scale* $N^{(2)} = N_a^{(2)*} + N_b^{(2)*} \neq N$. Assuming we have the optimal data quantity for domain $b$ becoming $m$ times than $N_b^*$, namely,

$$N_b^{(2)*} := m \cdot N_b^*$$

Then, from Eq. 4, the optimal data quantity for domain $a$ can be given as

$$\begin{aligned} N_a^{(2)*} &= \left[ \frac{\beta_a \gamma_a}{\beta_b \gamma_b} (N_b^{(2)*})^{\gamma_b + 1} \right]^{\frac{1}{\gamma_a + 1}} \\ &= \left[ \frac{\beta_a \gamma_a}{\beta_b \gamma_b} (m \cdot N_b^*)^{\gamma_b + 1} \right]^{\frac{1}{\gamma_a + 1}} \\ &= m^{\frac{\gamma_b + 1}{\gamma_a + 1}} \cdot \left[ \frac{\beta_a \gamma_a}{\beta_b \gamma_b} (N_b^*)^{\gamma_b + 1} \right]^{\frac{1}{\gamma_a + 1}} \\ &= m^{\frac{\gamma_b + 1}{\gamma_a + 1}} \cdot N_a^* \end{aligned} \tag{5}$$

It can be immediately seen that the optimal domain data $N_a^*$ and $N_b^*$ scale at different rates–new optimal data quantity for domain $a$ *does not* become $m$ times than before. This implies that the optimal data composition is scale-dependent and is different for different training data scales. This implies that the optimal data composition is scale-dependent and is different for different training data scales, establishing the main argument of this paper.

Since the ratio from Eq. 5, $(\gamma_b + 1)/(\gamma_a + 1)$, is constant and invariant to the change in the training data scale, it can be utilized to provide a direct approach for predicting the scaling of optimal data compositions–given as

$$N_a^{(2)*} = \left( \frac{N_b^{(2)*}}{N_b^*} \right)^{\frac{\gamma_b + 1}{\gamma_a + 1}} N_a^*$$

Equivalently, taking the logarithm for both sides of the equation, we have

$$\log N_a^{(2)*} = \log \left( \frac{N_b^{(2)*}}{N_b^*} \right)^{\frac{\gamma_b + 1}{\gamma_a + 1}} + \log N_a^*$$

Further, we show that one *does not* need to estimate any of the coefficients $(\gamma_a, \gamma_b)$ from the loss function to predict the optimal data quantity for each domain. Assume one have obtained the optimal data quantity for domains $a$ and $b$, $N_a^{(1)*}, N_b^{(1)*}$, at a *data scale* $N^{(1)} = N_a^{(1)*} + N_b^{(1)*}$ and the optimal data quantity $N_a^{(2)*}, N_b^{(2)*}$ at a *data scale* $N^{(2)} = N_a^{(2)*} + N_b^{(2)*}$ where $N^{(1)} \neq N^{(2)}$. This gives

$$\log N_a^{(2)*} = \frac{\gamma_b + 1}{\gamma_a + 1} \cdot (\log N_b^{(2)*} - \log N_b^{(1)*}) + \log N_a^{(1)*}$$

Then, for a different data scale where the optimal data quantity for domain $b$ is $N_b^{(3)*}$, the optimal data quantity for domain $a$ can be given as

$$\log N_a^{(3)*} = \frac{\gamma_b + 1}{\gamma_a + 1} \cdot (\log N_b^{(3)*} - \log N_b^{(2)*}) + \log N_a^{(2)*}$$

$$= \frac{\log N_a^{(2)*} - \log N_a^{(1)*}}{\log N_b^{(2)*} - \log N_b^{(1)*}} \cdot (\log N_b^{(3)*} - \log N_b^{(2)*}) + \log N_a^{(2)*}$$

W.l.o.g., consider $\frac{N_b^{(3)*}}{N_b^{(2)*}} = \frac{N_b^{(2)*}}{N_b^{(1)*}}$ where $\log N_b^{(2)*} - \log N_b^{(1)*} = \log N_b^{(3)*} - \log N_b^{(2)*}$, the equation above can be simplified to

$$\log N_a^{(3)*} = 2 \log N_a^{(2)*} - \log N_a^{(1)*}$$

and equivalently,

$$N_a^{(3)*} = \frac{(N_a^{(2)*})^2}{N_a^{(1)*}}$$

Defining compact representations $\mathbf{N}^{(i)*} = diag\{N_a^{(i)*}, N_b^{(i)*}\}$, the above results can be written as

$$\mathbf{N^{(3)*}} = \mathbf{N^{(2)*}}(\mathbf{N^{(1)*}})^{-1}\mathbf{N^{(2)*}}$$

which concludes the proof.

The process can be iterated (e.g., replacing $\mathbf{N^{(1)*}}$ or $\mathbf{N^{(2)*}}$ with $\mathbf{N^{(3)*}}$) to obtain optimal domain data quantity for different data scales. The example below provides a straightforward look on how this result can be operationalized. $\square$

**Remark 1** (An example). *This example helps visualize the operation pipeline.*

*If at training data scale $N^{(1)} = N_a^{(1)} + N_b^{(1)} = 200$, we have optimal domain data composition as $N_a^{(1)*} = 100, N_b^{(1)*} = 100$ (50% − 50%); and at scale $N^{(2)} = N_a^{(2)} + N_b^{(2)} = 500$, we have optimal domain data composition as $N_a^{(2)*} = 300, N_b^{(2)*} = 200$ (60% − 40%). Then, from the theorem, when the optimal domain data composition has $N_a^{(3)*} = (N_a^{(2)*})^2/N_a^{(1)*} = 900$, we can predict $N_b^{(3)*} = (N_b^{(2)*})^2/N_b^{(1)*} = 400$, which gives the optimal ratio at $N^{(3)} = N_a^{(3)} + N_b^{(3)} = 1300$ as 69% − 31%.*

*Similarly,*

*For $N_a^{(4)*} = 2700$, we have $N_b^{(4)*} = 800$, which gives the optimal ratio at $N^{(4)} = 3500$ as 77% − 23%*
*For $N_a^{(5)*} = 8100$, we have $N_b^{(5)*} = 1600$, which gives the optimal ratio at $N^{(5)} = 9700$ as 84% − 16%*
*For $N_a^{(6)*} = 24300$, we have $N_b^{(6)*} = 3200$, which gives the optimal ratio at $N^{(6)} = 27500$ as 88% − 12%*
*For $N_a^{(7)*} = 72900$, we have $N_b^{(7)*} = 6400$, which gives the optimal ratio at $N^{(7)} = 79300$ as 92% − 8%*
*For $N_a^{(8)*} = 218700$, we have $N_b^{(8)*} = 12800$, which gives the optimal ratio at $N^{(8)} = 231500$ as 94% − 6%*
*For $N_a^{(9)*} = 656100$, we have $N_b^{(9)*} = 25600$, which gives the optimal ratio at $N^{(9)} = 681700$ as 96% − 4%*

*We visualize it in Fig. 5.*

## D.2 SCALING LATENT SKILLS

We extend this theory to a general case where the evaluation loss is the perplexity averaged over training domains. Consider the evaluation is composed of a number of *independent* sub-tasks

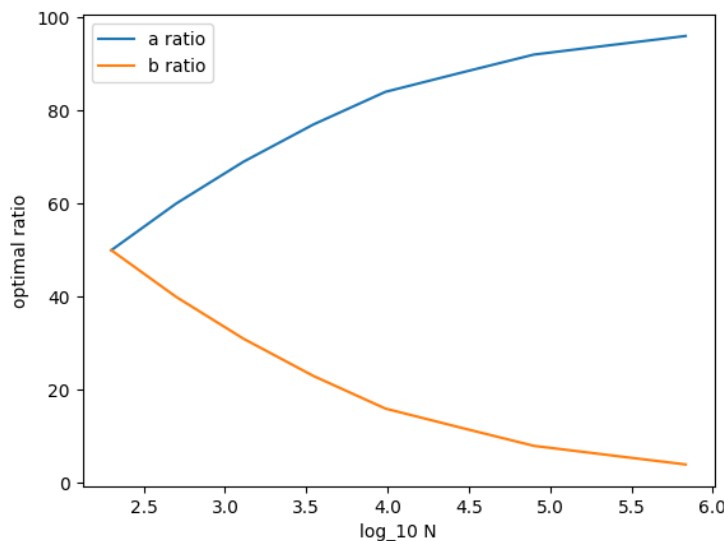

Figure 5: Illustration: optimal data composition scales in exponential-style functions with training data quantity.

("latent skills" (28)) which are hidden variables, where each of them observes a power law scaling law relationship with the amount of data contributing to this task ("equivalent data size"), $\mathcal{L} = \ell_0 + \beta_a \cdot K_a^{-\gamma_a} + \beta_b \cdot K_b^{-\gamma_b} + \beta_c \cdot K_c^{-\gamma_c} + \cdots$ where scalar $K_j \geq 0$ denote equivalent data size for $skill_j$, and constants $(\beta_j, \gamma_j) \geq 0$ are coefficients associated with $skill_j$, respectively. Mathematically, these latent skills can be seen as an orthogonal basis that spans the space of evaluation loss.

Consider training data from each domain $D_i$ contributes to these skills to varying degrees, where Equivalent data size for $skill_j$, $K_j$, is given as $K_j = c_{j,1} \cdot N_1 + c_{j,2} \cdot N_2 + \cdots$ where $N_i = w_i \cdot N$ denotes the amount of training data from domain $D_i$ and constant $c_{j,i}$ is the coefficient measuring the degree of contribution between domain $D_i$ and $skill_j$. Defining diagonal matrices for training data composition $\mathbf{N} = diag\{N_1, N_2, \cdots\}$ and skill data composition $\mathbf{K} = diag\{K_a, K_b, \cdots\}$, we have $\mathbf{K} = \mathbf{A}\mathbf{N}$, where $\mathbf{A}_{ji} = c_{j,i}$ is the matrix for coefficients. For simplicity, we consider training data from each domain will be *distributed* to the skills such that $\forall i, \sum_j N_i = 1$. This gives the amount of total training data from all domains is identical to the amount of total equivalent data for all skills, $\sum_j K_j = \sum_i N_i$. For a training data scale $N = \sum_i N_i = \sum_j K_j$, define optimal skill data composition $\mathbf{K}^* = diag\{K_a^*, K_b^*, \cdots\}$ as the minimizer of $\mathcal{L}$, given as $\mathbf{K}^* = \arg\min_{\sum_j K_j = N} \ell_0 + \beta_a \cdot K_a^{-\gamma_a} + \beta_b \cdot K_b^{-\gamma_b} + \cdots$. Theoretically, there can be an infinite number of latent skills. For analysis, we consider a finite number of $k$ independent skills *most important* for the evaluation. This can considered as performing Principal Components Analysis (PCA) with orthogonal transformation and selecting the first $k$ independent components. We consider the standard scenario with an equal number of relevant skills and data domains where $k = m$ and $\mathbf{A}$ is a square matrix with full rank. This describes the case where this optimization problem is well-defined. We discuss in App. D.2 what will happen in other scenarios. In this case, $\mathbf{A}$ is invertible and the corresponding optimal training data composition for $\mathbf{K}^*$ can be given as $\mathbf{N}^* = \mathbf{A}^{-1}\mathbf{K}^*$.

We provide the following theorem, which states that for the scenario described above, optimal training data composition scales in exponential-style functions with training data quantity and can be directly predictable from that of smaller scales *without needing to identify the latent skills*.

**Theorem 2** (Scaling Latent Skills). *Consider the evaluation is composed of a number of independent sub-tasks ("latent skills") where each of them observes a power law scaling law relationship with the amount of data contributing to this task ("equivalent data size"). Namely, $\mathcal{L} = \ell_0 + \beta_a \cdot K_a^{-\gamma_a} + \beta_b \cdot K_b^{-\gamma_b} + \beta_c \cdot K_c^{-\gamma_c} + \cdots$ where scalar $K_j \geq 0$ denote equivalent data size for $skill_j$, and constants $(\beta_j, \gamma_j) \geq 0$ are coefficients associated with $skill_j$, respectively. Define diagonal matrices for training data composition $\mathbf{N} = diag\{N_1, N_2, \cdots\}$ and skill data composition $\mathbf{K} = diag\{K_a, K_b, \cdots\}$. Consider training data from each domain $D_i$ contributes to these skills to varying degrees, given as $\mathbf{K} = \mathbf{A}\mathbf{N}$ where $\mathbf{A}_{ji} = c_{j,i}$ is the matrix*

*for coefficients. Assume the amount of total training data from all domains is identical to the amount of total equivalent data for all skills, $\sum_j K_j = \sum_i N_i$. Assume there is a finite number of latent skills and data domains and $\mathbf{A}$ is a square matrix with full rank. For a training data scale $N = \sum_i N_i = \sum_j K_j$, define optimal skill data composition $\mathbf{K}^* = diag\{K_a^*, K_b^*, \cdots\}$ as the minimizer of $\mathcal{L}$ s.t. $\sum_j K_j = N$ with corresponding optimal training data composition If we have optimal data compositions $\mathbf{N}^{(1)*} = diag\{N_a^{(1)*}, N_b^{(1)*}, \cdots\}$ where its corresponding skill data composition $\mathbf{K}^{(1)*} = diag\{K_a^{(1)*}, K_b^{(1)*}, \cdots\} = \mathbf{A}\mathbf{N}^{(1)*}$ minimizes $\mathcal{L}$ s.t. $\sum_j K_j = \sum_i N^{(1)*} = N^{(1)}$, and $\mathbf{N}^{(2)*} = diag\{N_a^{(2)*}, N_b^{(2)*}, ...\}$ where its corresponding skill data composition $\mathbf{K}^{(2)*} = diag\{K_a^{(2)*}, K_b^{(2)*}, ...\} = \mathbf{A}\mathbf{N}^{(2)*}$ minimizes $\mathcal{L}$ s.t. $\sum_j K_j^{(2)*} = \sum_i N^{(2)*} = N^{(2)}$ where $N^{(2)} \neq N^{(1)}$, then, other optimal data compositions $\mathbf{N}^{(3)*} = diag\{N_a^{(3)*}, N_b^{(3)*}, ...\}$ where its corresponding skill data composition $\mathbf{K}^{(3)*} = diag\{K_a^{(3)*}, K_b^{(3)*}, \cdots\} = \mathbf{A}\mathbf{N}^{(3)*}$ minimizes $\mathcal{L}$ s.t. $\sum_j K_j^{(3)*} = \sum_i N^{(3)*} = N^{(3)}$ where $N^{(3)} \neq N^{(2)} \neq N^{(1)}$ can be given as $\mathbf{N}^{(3)*} = \mathbf{N}^{(2)*}(\mathbf{N}^{(1)*})^{-1}\mathbf{N}^{(2)*}$.*

*Proof.* By definition, we have

$$\mathbf{A}\mathbf{N}^{(1)*} = \mathbf{K}^{(1)*}, \quad \mathbf{A}\mathbf{N}^{(2)*} = \mathbf{K}^{(2)*}, \quad \mathbf{A}\mathbf{N}^{(3)*} = \mathbf{K}^{(3)*}$$

From results of Theorem 1, we have

$$\mathbf{K}^{(3)*} = \mathbf{K}^{(2)*}(\mathbf{K}^{(1)*})^{-1}\mathbf{K}^{(2)*}$$

which gives

$$\mathbf{A}\mathbf{N}^{(3)*} = (\mathbf{A}\mathbf{N}^{(2)*})(\mathbf{A}\mathbf{N}^{(1)*})^{-1}\mathbf{A}\mathbf{N}^{(2)*}$$

Since $\mathbf{A}$ is invertible and $\mathbf{N}$ and $\mathbf{K}$ are diagonal matrices, naturally,

$$(\mathbf{A}\mathbf{N}^{(1)*})^{-1} = (\mathbf{N}^{(1)*})^{-1}\mathbf{A}^{-1}$$

and we have

$$\mathbf{A}\mathbf{N}^{(3)*} = \mathbf{A}\mathbf{N}^{(2)*}[(\mathbf{N}^{(1)*})^{-1}\mathbf{A}^{-1}]\mathbf{A}\mathbf{N}^{(2)*} = \mathbf{A}\mathbf{N}^{(2)*}(\mathbf{N}^{(1)*})^{-1}\mathbf{N}^{(2)*}$$

This directly gives

$$\mathbf{N}^{(3)*} = \mathbf{A}^{-1}\mathbf{A}\mathbf{N}^{(2)*}(\mathbf{N}^{(1)*})^{-1}\mathbf{N}^{(2)*} = \mathbf{N}^{(2)*}(\mathbf{N}^{(1)*})^{-1}\mathbf{N}^{(2)*}$$

which completes the proof.

The above result does not require identifying the latent skills or observing skill data compositions $\mathbf{K}$. Rather, the theorem gives that as long as the coefficient matrix $\mathbf{A}$ is invertible, the scaling of $\mathbf{N}$ complies to the same scaling law as in Sec. 3.3. $\qquad\square$

**Remark 2** (what happens when $\mathbf{A}$ is not invertible.)**.** *In general, if $\mathbf{A}$ is not invertible, scaling for optimal training data composition is not directly predictable. Specifically, if $\mathbf{A}$ does not have full rank, there exists redundant domains/data sources where their contribution to the skills are identical/exact multipliers of each other. Some data sources may not be needed at any scale; if $\mathbf{A}$ has more rows than columns (more domains than skills), this suggests multiple training data compositions can achieve the same skills data composition and the optimal training data compositions are non-unique (infinitely many). If $\mathbf{A}$ has more columns than rows (more skills than domains), this means there are too many skills to optimize for. No optimal training data composition exists and one has to make trade-offs. If this is relevant to the practical needs, training data may be processed with additional techniques such as clustering and split into more different domains.*

## APPENDIX E   EXPERIMENTAL DETAILS AND ADDITIONAL RESULTS

### E.1   EXPERIMENTAL DETAILS OF SEC. 5.2

**Model Training**   `GPT-2 Large` is a variant of the `GPT-2` architecture, featuring an embedding dimension of 1280, 36 transformer layers, and 20 attention heads. We rely on the Hugging Face Transformers library for implementation (72). Specific training hyperparameters are detailed in Table 2.

| | |
|---|---|
| Architecture | gpt2 |
| Optimizer | AdamW |
| Tokenizer Vocabulary Size | 50257 |
| Batch Size Per Device | 1 |
| Gradient Accumulation Steps | 10 |
| Maximum Learning Rate | 2e-4 |
| LR Schedule | Linear |
| Weight Decay | 1e-2 |
| Warm-up Ratio | 10% |
| Epochs | 3 |
| GPU Hardware | 8x NVIDIA A100/8x NVIDIA H100 |

Table 2: The list of hyperparameters for `GPT-2 Large` pretraining.

**Dataset Details**   The `RedPajama` dataset is available at: `https://huggingface.co/datasets/togethercomputer/RedPajama-Data-1T`. The 7 domains involved are characterized as follows:

- **`Commoncrawl`**: A vast repository of web-crawled data, providing a heterogeneous mix of internet text.
- **`C4`**: The Colossal Clean Crawled Corpus, filtered to remove low-quality content, thus ensuring the reliability and cleanliness of the data.
- **`GitHub`**: This domain includes a compilation of publicly available code repositories, offering a rich source of syntactic and semantic patterns inherent in programming languages.
- **`Books`**: A collection of textual content from published books, providing diverse narrative styles and complex character developments.
- **`ArXiv`**: Comprising scientific papers primarily from the fields of physics, mathematics, computer science, and quantitative biology, this domain offers high-quality, scholarly content.
- **`Wikipedia`**: A well-organized and meticulously curated dataset of encyclopedia articles, delivering a broad spectrum of knowledge across multiple disciplines. We only use English samples with 'en' in meta-data.
- **`StackExchange`**: This domain captures a variety of user-generated content from discussions and question-answer sessions across numerous technical topics.

Given copyright restrictions with the `Books` domain on Hugging Face, we have opted for an alternative source available at `https://yknzhu.wixsite.com/mbweb`.

For each domain, we ensure only samples with more than 1000 characters are retained. For each sample, the first 1000 characters are truncated, with the exception of the `ArXiv` and `GitHub` domains where we randomly extract a continuous block of 1000 characters. For the `Wikipedia` domain, we keep only those samples that are in English. Samples are selected without replacement, based on the computed data volume for each domain. Additionally, for each domain, a held-out dataset comprising 10K samples is reserved to evaluate the perplexity of the pretrained model.

E.2   EXPERIMENTAL DETAILS OF SEC. 5.3

**Model Training**   We employ the `BERT-base-uncased` model from the Hugging Face Transformers library. Originally, `BERT`'s pretraining scheme involved MLM and next sentence prediction (NSP); however, in our experiments, we exclusively utilize MLM. Detailed training hyperparameters can be found in Table 3.

**Dataset Details**   The 5 domains of training data utilized are listed as follows:

- **`Amazon Reviews`**: A compilation of customer reviews from Amazon, widely utilized in sentiment analysis studies, available at: `https://huggingface.co/datasets/amazon_us_reviews`.

| | |
|---|---|
| Architecture | bert-base-uncased |
| Max Token Length | 300 |
| Mask Token Percentage | 15% |
| Optimizer | AdamW |
| Batch Size Per Device | 12 |
| Devices | 4 |
| Maximum Learning Rate | 1e-4 |
| LR Schedule | Linear |
| Weight Decay | 1e-2 |
| Warm-up Steps | 3000 |
| Epochs | $1 \sim 4$ |
| GPU Hardware | 4x NVIDIA RTX A6000 |

Table 3: The list of hyperparameters for `BERT` pretraining.

- **`Arxiv`**: Comprises 1.7 million articles from arXiv, available at: `https://www.tensorflow.org/datasets/catalog/scientific_papers`.

- **`Books`**: A corpus of 11,038 novels by unpublished authors across 16 genres, available at: `https://yknzhu.wixsite.com/mbweb`.

- **`Wikipedia`**: Offers datasets extracted from Wikipedia in various languages, available at: `https://www.tensorflow.org/datasets/catalog/wikipedia`. We only use English samples with 'en' in meta-data.

- **`Open WebText Corpus (OWTC)`**: A corpus of English web texts from Reddit posts, available at: `https://skylion007.github.io/OpenWebTextCorpus/`.

3 held-out non-training domains used in the evaluation include:

- **`Pubmed`**: Features 19,717 diabetes-related publications from the PubMed database, organized into three classes and linked by a network of 44,338 citations, available at: `https://www.tensorflow.org/datasets/catalog/scientific_papers`

- **`News`**: Comprises a significant collection of news articles derived from `CommonCrawl`, specifically from 5000 news domains indexed by Google News, available at: `https://github.com/rowanz/grover/blob/master/realnews/README.md`

- **`GitHub`**: A curated selection from the `RedPajama` dataset, this segment includes an array of open-source code projects, available at: `https://huggingface.co/datasets/togethercomputer/RedPajama-Data-1T`

### E.3 IMPLEMENTATION DETAILS OF BASELINES

**Implementation details** We followed the official implementation[3] of DOREMI for our experiments. We evaluated two sets of reference domain weights: (1) the domain weights utilized in the LLaMA-2 paper (11) (referred to as LLaMA weights), and (2) uniform weights. Both the reference and proxy models have 120M parameters and are trained from scratch. We use `GPT-2` tokenizer with a vocabulary size of roughly 50K. For LLaMA weights, we train each model for 20K, 50K and 200K steps for comparison. For uniform weights, we train each model for 10K, 20K and 50K steps. Refer to Table 4 for detailed hyperparameters. The effect of reference weights on the output DOREMI is discussed in Fig.9.

### E.4 EVALUATION DETAILS

**GPT/CLM** The following tasks are considered for downstream performance evaluation, in line with the setup from (10; 21). For few-shot tasks, the demonstrations are sampled at random.

- **`BoolQ`** (33) consists of a question-answering format that requires binary yes/no answers.

---

[3]`https://github.com/sangmichaelxie/doremi`

| Architecture | Decoder-only LM |
|---|---|
| Max Token Length | 1024 |
| Optimizer | AdamW |
| Batch Size Per Device | 8 |
| Devices | 8 |
| Maximum Learning Rate | 2e-4 |
| LR Schedule | Linear |
| Weight Decay | 1e-2 |
| Warm-up Steps | 3000 |
| Epochs | 1 |
| GPU Hardware | 8x NVIDIA RTX A6000 |

Table 4: The list of hyperparameters for DOREMI.

- **HellaSwag** (34) challenges models on their ability to make commonsense inferences.
- **PIQA** (35) focuses on evaluating a model's commonsense reasoning regarding physical interactions.
- **TruthfulQA** (36) is designed to assess the ability of models to generate truthful and factual responses.
- **PubMedQA** (37) offers a dataset for evaluating question-answering in the biomedical domain.
- **CrowsPairs-English** (38) tests models on their ability to identify and correct stereotypical biases in English text.
- **ARC-Easy** (39) presents a set of relatively simpler scientific reasoning questions, aimed at evaluating a model's basic understanding of scientific principles.
- **BigBench-Novel Concepts** (40) serves as a test of the model's creative abstraction skills, challenging it to make sense of scenarios that it could not have memorized during training.

**BERT/MLM**  For each task, we conduct supervised fine-tuning on the corresponding training data and test the fine-tuned model on the validation data. The hyperparameters for supervised fine-tuning are given in Table 5.

| Architecture | bert-base-uncased |
|---|---|
| Max Token Length | 128 |
| Batch Size Per Device | 8 or 300 |
| Optimizer | AdamW |
| Devices | 4 |
| Maximum Learning Rate | 2e-5 or 5e-5 |
| Epochs | 3 |
| GPU Hardware | 4x NVIDIA RTX A6000 |

Table 5: The list of hyperparameters for supervised fine-tuning of BERT.

### E.5 ADDITIONAL RESULTS OF SEC. 5.2

Fig. 6 depicts AUTOSCALE-predicted domain weights for training 774M Decoder-only LMs. Optimal data quantity for each domain grows in exponential-style functions with training data scale (left) where data sources with diverse samples (e.g., C4) are upweighted relative to domains with standard format (e.g., Wikipedia).

Fig. 7 shows that when training on up to 5B tokens, AUTOSCALE-predicted weights decreases val loss at least 25% faster than any baseline with up to 37% speed up.

Fig. 8 visualizes domain weights used for training GPT-2 Large, given by different methods.

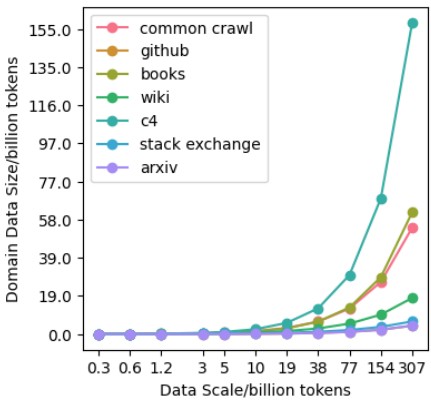 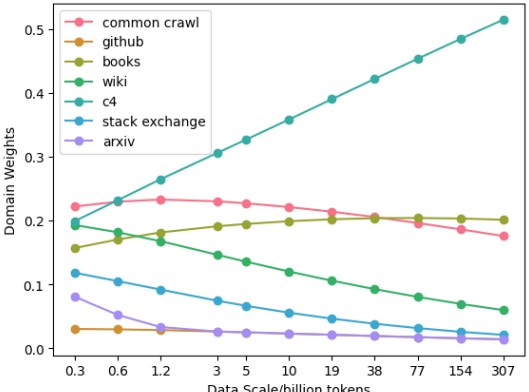

(a) AUTOSCALE-predicted optimal data quantity for each domain as training data scales up.

(b) AUTOSCALE-predicted optimal domain weights as training data scales up.

Figure 6: AUTOSCALE-predicted domain weights for training 774M Decoder-only LMs. Optimal data quantity for each domain grows in exponential-style functions with training data scale (left) where data sources with diverse samples (e.g., `C4`) are upweighted relative to domains with standard format (e.g., `Wikipedia`).

Table 6 examines the domain-specific perplexity of `GPT-2 Large` trained on 3 billion tokens, respectively. Notably, AUTOSCALE achieves the lowest average validation perplexity and significantly reduces the perplexity in the worst-performing domains.

Fig. 9 visualizes DOREMI optimized domain weights with different reference weights and training steps. Training proxy/reference models for different steps gives different weights. It is unclear which weights are optimal. DOREMI recommends 200k steps, which equals >100B tokens in the default setup. Since optimization was conducted relative to the reference weights, reference weights have a profound impact on DOREMI's output.

| **Domain/Method** | AutoScale | DoReMi (Ref) | Data Mixing Laws (ref) | LLaMA | Uniform (30% more tokens) |
|---|---|---|---|---|---|
| Common Crawl | 25.598 | **24.116** | 30.824 | 21.464 | 28.351 |
| Github | 7.482 | 6.678 | **5.845** | 7.376 | 5.784 |
| Books | **29.162** | 33.324 | 34.450 | 35.533 | 31.14 |
| Wikipedia | 18.828 | **17.154** | 26.795 | 21.110 | 19.57 |
| C4 | **34.242** | 39.429 | 38.521 | 37.393 | 40.323 |
| Stack Exchange | 15.991 | 15.393 | **14.519** | 20.133 | 13.890 |
| Arxiv | 16.558 | 15.638 | **12.372** | 17.598 | 13.082 |
| **Average** | **21.123** | 21.676 | 23.333 | 22.944 | 21.736 |
| **Worst-domain** | **34.242** | 39.429 | 38.521 | 37.393 | 40.323 |

Table 6: Domain perplexity for 774M Decoder-only LMs trained for 3B tokens. AUTOSCALE notably achieves the lowest average validation perplexity while also significantly decreasing worse-domain perplexity.

### E.6    ADDITIONAL RESULTS OF SEC. 5.3

Fig. 10(b) shows the results on fitting validation loss with power-law functions, directly approximating how loss changes with each domain's data quantity. Compared to `BERT` models trained with MLM (right), `GPT` models trained with CLM (left) demonstrate a much stronger response to domain reweighting. In final results, `GPT`/CLM achieved $> 2\times$ speed-up margins relative to uniform weights compared to `BERT`/MLM.

Fig. 11 depicts the AUTOSCALE-predicted domain weights for training `BERT`. It is evident that optimal data quantity for each domain grows in exponential-style functions with training data scale

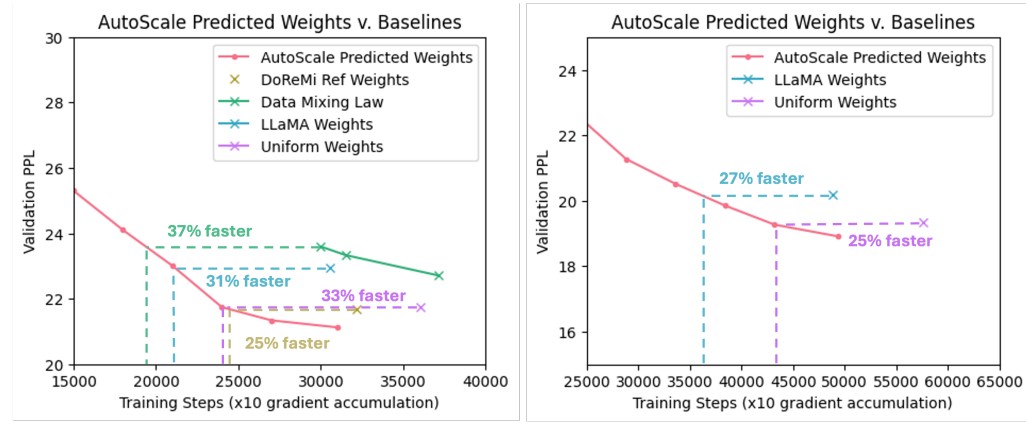

(a) Training Decoder-only LMs for 3B tokens.     (b) Training Decoder-only LMs for 5B tokens.

Figure 7: AUTOSCALE-predicted weights decreases val loss at least $25\%$ faster than any baseline with up to $37\%$ speed up. Despite LLaMa weights being very different from uniform weights, they yield highly similar training efficiency at these data scales.

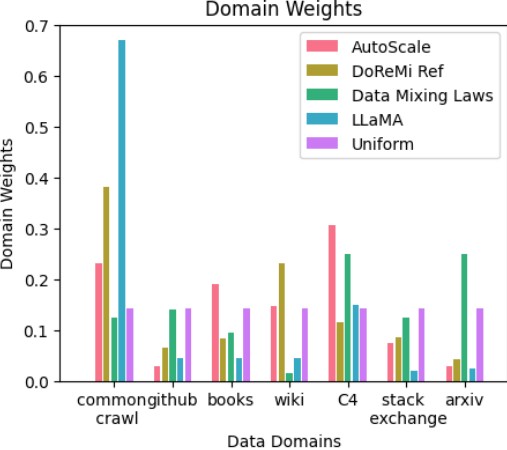

Figure 8: Domain Weights used for training 774M Decoder-only LMs for 3B tokens. (Domain weights for DATA MIXING LAWS and DOREMI are from references (6) and (5), respectively, which are implemented on the same datasets/data domains with highly similar model architecture/model size/tokenizers.)

where data sources with diverse samples (e.g., WebText) are upweighted relative to domains with standard format (e.g., ArXiv).

Table 7 shows AUTOSCALE notably improving training efficiency for BERT models on all scales–even for a considerably large scale, 288k steps, the speedup margin remains visible.

| Data Scale/steps | 18k | 36k | 72k | 144k | 288k |
|---|---|---|---|---|---|
| Final Loss (exp) | 38.32 | 16.94 | 10.97 | 8.13 | 6.30 |
| Steps Saved | 5k (28%) | 5k (14%) | 10k (14%) | 20k (14%) | 20k (10%) |

Table 7: AUTOSCALE notably improving training efficiency for BERT models on all scales–even for a considerably large scale, 288k steps, the speedup margin remains visible.

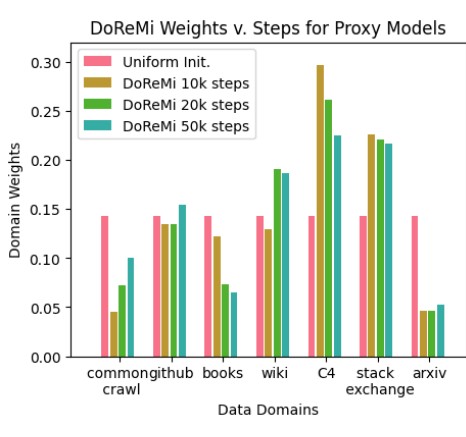
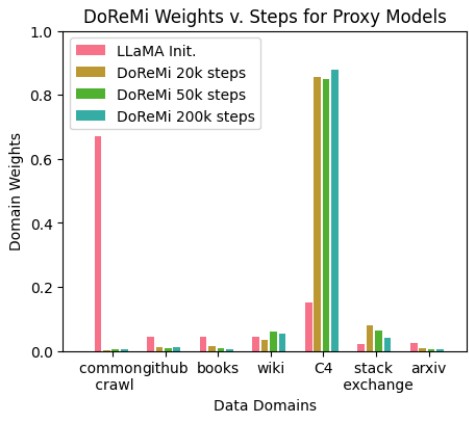

(a) with Uniform Reference Weights                    (b) with LLaMA Reference Weights (Default)

Figure 9: DOREMI with different reference weights and steps. Training proxy/reference models for different steps gives different weights. It is unclear which weights are optimal. DOREMI recommends 200k steps, which equals >100B tokens in the default setup. Since optimization was conducted relative to the reference weights, reference weights have a profound impact on DOREMI's output.

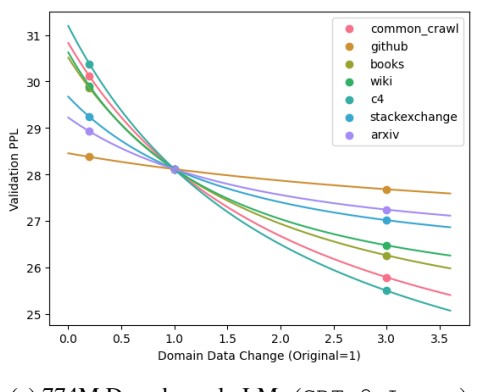
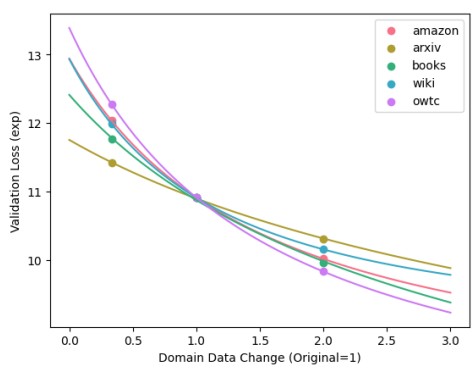

(a) 774M Decoder-only LMs (GPT-2 Large)              (b) Encoder-only LMs (BERT-case)

Figure 10: Fitting validation loss with power-law functions, directly approximating how loss changes with each domain's data quantity. Compared to BERT models trained with MLM (right), GPT models trained with CLM (left) demonstrate a much stronger response to domain reweighting. In final results, GPT/CLM achieved $> 2\times$ speed-up margins relative to uniform weights compared to BERT/MLM.

### E.7 RUNTIME ANALYSIS

Training a GPT-2 large model from scratch for 3B tokens requires 15.5 hours on 8x NVIDIA A100 40GB SXM GPUs or 9 hours on 8x NVIDIA H100 80GB GPUs. Training time increases linearly with the number of training tokens on both types of GPUs.

Training BERT-base models takes 2 hours for every 18k steps on 4x NVIDIA A6000 48GB GPUs. Computational time grows linearly with the number of training steps.

Training reference models for DOREMI takes one hour for every 10K steps on 8x NVIDIA A6000 48GB GPUs. Computational time grows linearly with the number of training steps. Similar runtime for training proxy models for DOREMI.

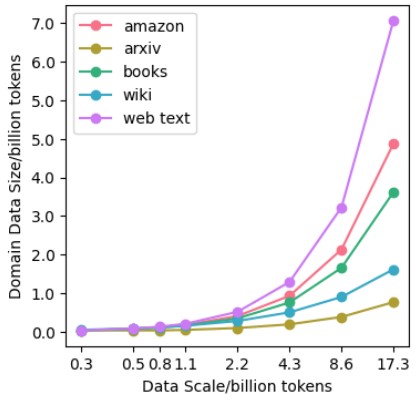

(a) AUTOSCALE-predicted optimal data quantity for each domain as training data scales up.

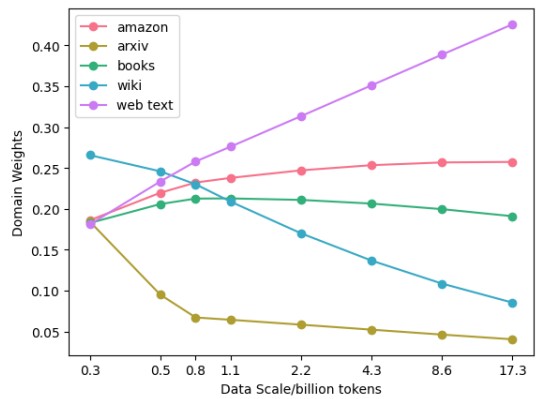

(b) AUTOSCALE-predicted optimal domain weights as training data scales up.

Figure 11: AUTOSCALE-predicted domain weights for training Encoder-only LMs (BERT). Optimal data quantity for each domain grows in exponential-style functions with training data scale (left) where data sources with diverse samples (e.g., WebText) are upweighted relative to domains with standard format (e.g., ArXiv).

