# OpenReview forum: "AutoScale: Automatic Prediction of Compute-optimal Data Compositions for Training LLMs"
_ICLR.cc/2025/Conference — Submitted to ICLR 2025_

### Official Review · Reviewer_Ad96 · 2024-11-01

**Soundness:** 3
**Presentation:** 3
**Contribution:** 3
**Rating:** 5
**Confidence:** 3

**Summary:**

This work proposes a method called “AutoScale” that helps predict the optimal composition of pre-training data for LLMs. It challenges the conventional notion of determining this via small scale experiments and simply applying them to a large scale where two axes change (data scale, parameter count). The experiments show a very promising line of research and it was a pleasure to read.

I couldn’t check the math as well as I would have liked to.

**Strengths:**

* Very strong work in terms of the hypothesis and experimental setup albeit at smaller scales. The promise of finding optimal weights for training large networks without having to guesstimate it is a very attractive proposition.
* The plots are really well done. They drive the main idea of the paper very well(especially Fig 1 (a, e) )

**Weaknesses:**

I would like to list the following weakness fully ensuring the authors that I am not unreasonable and am completely open to increasing my score if these are addressed/answered satisfactorily.
* The work proposes using a different approach to finding optimal data weights for a given pre-training compute budget. This is well explained via results but does in fact require training the original size model. Given that we obtain suboptimal performance via the conventional way( smaller model, fewer data), an analysis showing how much performance could be gained by spending the compute and training these (equal parameter) networks would be useful.
* For Takeaway 1, Fig 1(b) only has 2 data points. Additional points would help make the case stronger. It’s a tough sell to make such a bold statement with two data points. But I’m hoping I am wrong :)
* Maybe I missed this, but the repeated claims that Wikipedia should be sampled less at a higher scale is a result of the OLS fit. But no experiment actually confirmed this fact in the paper, right ? Since the max scale was 1.2B ? Please correct me if I’m wrong.

General Comments/Typos:
* [Section2] :  “this work contribute”  -> “this work contributes”
* [Section 3.1] : wi = Ni/N => wi = Si/N ?
* [Algorithm 1] : Train the model on data S = ({S1 . . . Sm} \ Si) => S = ({S1 . . . Sm} \ Sj) ?
* Some of the font sizes are very distracting to read.

**Questions:**

* Even at a smaller scale, I see opportunities of clear promise where we could have had more points between 0.3B and 1.2B and show some trend. Any specific reason this was not done/ increased to more than 1.2B ? With scale, a lot of problems disappear that are apparent at lower scales.

---

> ### Author Response · Authors · 2024-11-23
> **Rebuttal period ends in a few days–we anticipate your feedback!**
>
> Dear Reviewer Ad96,
>
> As the rebuttal/author discussion period ends in a few days, we sincerely look forward to your feedback. The authors are deeply appreciative of your valuable time and efforts spent reviewing this paper and helping us improve it.
>
> It would be very much appreciated if you could once again help review our responses and additional results and let us know if these address or partially address your concerns and if our explanations are heading in the right direction :)
>
> Please also let us know if there are any further questions or comments about this paper. We strive to consistently improve the paper and it would be our pleasure to have your precious feedback!
>
> Kind Regards,\
> Authors of Submission1690

---

> > ### Comment · Reviewer_Ad96 · 2024-11-25
> >
> > Could the author's paste their response to each of the questions ? It seems you have answered Reviewer Qo6P's comments in this comment, but not all of mine,

---

### Official Review · Reviewer_JgHF · 2024-11-04

**Soundness:** 2
**Presentation:** 2
**Contribution:** 2
**Rating:** 3
**Confidence:** 2

**Summary:**

This work presents a method to estimate optimal data domain weights at large training-data scales by extrapolating via exponential functions fit to smaller-scale training runs. The proposed method is evaluated on GRT-2 Large + RedPajama and BERT pretraining, and compared to extant method baselines.

**Strengths:**

The general problem the authors attempt to address is important, and the assessment that present methods are limited and that performance headroom is available is well-framed. The code is open-sourced. The evaluations presented are limited, but positive.

**Weaknesses:**

In general, the writing is difficult to parse. It is frequently frustratingly vague, including in the Abstract, where the actual method is alluded to but not elucidated. In the actual methods section, important questions about the method are unanswered, leaving the method underspecified (is the learning rate schedule (linear? presumably linear decay?) the same for the tuning runs as for the final run? Is the decay timing adjusted to the compute budget? The value of the final validation loss hinges critically on this - yet it goes unmentioned). There is no addressing of the profound difficulties this method (and others like it) can be expected to have around epochs for individual datasets. A more thorough analysis would identify and investigate this issue with experiments demonstrating specific datasets being sampled for > 1 epoch, and the subsequent breakdown of the "scaling law" prediction. Evalutation is purely comparative to other methods, and does not assess to what extent the predicted 'optimal' values might differ from more expensively traditionally-derived 'optimal' values. No discussion of the relative cost of the method (with its 'linearly scaling' cost in the number of datasets) is mentioned, though it is clear it would become prohibitively expensive for dataset mixtures with more than a handful of individual datasets. The method proposed is prohibitively expensive at large model sizes, and seems unlikely to scale to larger compute budgets even at small dataset sizes due to the issue of datasets passing through multiple epochs, which is unaddressed in this work. This limitation goes unmentioned.

**Questions:**

Fig 1: [a] suggests that you've tuned 6 models between 30M and 1.2 B tokens, yet [c] shows only three models being used to fit the predictor model. Why is that? where are the other data points? And are *all* of the linear fits R2=0.998? Is that the average R2? Also, [d] shows the predictions of the model extrapolated past 1.2 B to 307 B? Why are you not showing the training data points (between 30M and 1.2 B) as well? And isn't the largest model you look at trained to 10B? why show this extrapolation to so far beyond where you explore? This seems misleading. The x-axis should say (log scale) as well. In [b] the color used for the 1.2B model is the color used for the 0.6B model in [a]. And there is a typo in the title ('scale - depedent' -> "dependent"). In [e] the 38% improvement looks to be overstated due to the noise of those evaluation curves, you could just as easily pick out the peak in Autoscale curve at step 86k and the point in the Uniform curve at step 100k to get a smaller improvment result with the same underlying data.

Table 1: boolq has the Autoscale value bolded as 'best' but the Data Mixing Laws value is greater. Also, consider place your method on the bottom row separated by a thin line.

Fig2: What is being depicted here? Is this showing power laws being fit to 3 empirical datapoints? Is the first column of points supposed to be at 0? It looks like the points are at [0.2, 1, 3] on the x-axis?


Nits:

Throughout: "AutoScale" is consistently the wrong font size. Please fix. Similarly, in section 5.2 the font size of the methods needs to be fixed. And in line 418 'from' is included in the method font instead of the default font.
181: work contribute -> contributes
379: N^(1)* is missing the N in summation
465: much lowered -> much lower
155 'a consistent shift can be observed', please be more specific, what is shifting, how is it consistent?

---

> ### Author Response · Authors · 2024-11-20
> **Response by Authors (1/3)**
>
> We thank the reviewer for the review and appreciate the feedback. Here we provide explanations for the listed weaknesses and responses to the questions.
>
> ---
>
> ## 1. Experiment setup
>
> ### a. Learning rate schedule
>
> This work follows the standard setup for training GPT-2-style LMs with a linear learning rate schedule and a 10% warmup ratio. The learning rate schedule and warmup steps are relative to the total steps in each training run, automatically scaled for each experiment. We documented the hyperparameters for pre-training GPT-2 Large models in Table 2.
>
> ### b. Number of epochs for different domains
>
> We agree with the argument that sampling different domains with different numbers of epochs could lead to misaligned results and confusing comparisons. **Thus, in this work, we strictly controlled this factor to ensure in every training run, regardless of its mixture, we train on data from each domain with the same number of epochs.** The source dataset used in this work, RedPajama, is large enough so we are always able to obtain the target amount of non-repeating samples for every data mixture.
>
> ### c. Comparative evaluation
>
> To the best of our knowledge, **there are no existing solutions for determining the ground truth optimal domain weights, except by computationally intractable exhaustive grid search with training the full model on all combinations of domain weights.** *[DoReMi: Optimizing Data Mixtures Speeds Up Language Model Pretraining, 2023]* is one of the earliest works on optimizing training data mixture for pre-training LLMs and has been widely referred to as the main baseline in subsequent works. Thus, we formulate the data optimization problem in Section 3.1 as a bilevel optimization problem where no direct solution is computationally tractable. Consequently, we developed the Direct Data Optimization (DDO) algorithm in Section 3.2 which leverages scaling law analysis to provide a global approximation to the original optimization problem, **achieving high-accuracy optimization results with plausible computation complexity. This already pushed forward the frontier of solving the original data optimization problem.** But still, we do not settle for a solution that requires training the model with a full data budget multiple times. This led to the development of the novel AutoScale tool, which automatically predicts optimal training data compositions at larger scales based on compositions optimized at smaller scales.

---

> ### Author Response · Authors · 2024-11-20
> **Response by Authors (2/3)**
>
> ### D. Computation overhead
>
> DDO requires training $2M +1$ models where $M$ is the number of data domains. This is conducted on a data scale **magnitudes smaller** than the target compute budget. AutoScale requires obtaining optimized domain weights with DDO on two different data scales. This allows for predicting the domain weights for training the target model with a much larger data budget. In comparison, the full data budget for 1B-parameter models is 2~3T tokens** (ref: 1.5T tokens in *[OpenELM: An Efficient Language Model Family with Open Training and Inference Framework]*, 2T tokens in *[TinyLlama: An Open-Source Small Language Model]*, 3T tokens in *[olmo: Accelerating the science of language models]*) **where a 0.5% speed-up would justify the cost.** **We are adding this discussion to the manuscript and the limitations.**
>
> *Besides, the computation efficiency for AutoScale is advantageous relative to existing works.** In the default setup, in each iteration, DoReMi requires training two models each with 200k steps with token_length 1024 and batch_size 64*8, **which equals >100B tokens and ~10x than AutoScale.** This process needs to iterate multiple times, resulting in a multitude of compute. This has not included the computation for conducting the grid search to determine which proxy model to use. *[Rephrasing the Web: A Recipe for Compute and Data-Efficient Language Modeling]* explores improving LLM pretaining efficiency via training on synthetic texts generated by pre-trained LLMs. **The computation overhead for generating synthetic texts with 7B-parameter models is in the same magnitude as training the target 1B-parameter model**.
>
> Further, as listed in the discussion with Reviewer mxrW, even though the cost of training full-size models on a small proportion of its target data budget should be within a low percentage of the total training cost, **there's room for further improvements.** The next step could be **applying the scaling law solution developed for different data budgets to different model sizes and directly predicting optimal data composition across model scales.** The original paper on scaling laws reports that model performance is empirically predictable with power law functions w.r.t. to either the training data size OR the model size–these two were treated as equal, independent factors. Theoretically, the methodology we developed in this work could be applied to scaling with the model size as well. This extension could be an independent work parallel to this manuscript. Should the results turn out to be positive, then the next step will be combining these two methods into a unified framework to predict optimal data mixtures for different compute budgets and model sizes at the same time. **We are listing this in future directions and calling for contributions from the community.**

---

> > ### Author Response · Authors · 2024-11-20
> > **Response by Authors (3/3)**
> >
> > ## Questions: Linear fitting in Figure 1[a]
> >
> > The reported $R^2=0.998$ is the averge from all curves. In this work, we fit the proposed AutoScale predictor with optimal domain weights obtained at 0.3B~1.2B data budgets (where only 2 out 3 datapoints are needed for fitting AutoScale). Data budgets smaller than 0.1B tokens are out of the normal range and only provided for qualitative analysis. Due to data sampling and stochasticity from the ML training pipeline, the possible noise in small-scale results may outweigh the benefits of adding these results to quantitative analysis.
> >
> > ---
> >
> > ## Question: Extrapolation on larger data scales
> >
> > AutoScale is designed to predict optimal domain weights on larger data scales where the direction solution to the data optimization problem cannot be obtained. It is intended used for AutoScale to provide forecasts on optimal data mixtures before the model is trained. Though the intervals on the X-axis are not uniform, the values on the X-axis are the actual data budgets (in billion tokens).
> >
> > ---
> >
> > ## Question: Relative improvements on non-monotonic curves
> >
> > We agree that the relative improvements on non-monotonic performance curves have certain ambiguity. In this work, we followed the same practice as in representing works on related problems such as [DoReMi] and [Rephrasing the Web: A Recipe for Compute & Data-Efficient Language Modeling, 2024]. It appears the common practice to define relative performance improvements in this manner. Following the same procedure could allow easier comparison with other results.
> >
> > ---
> >
> > ## Question: X-axis in Figure 2
> >
> > Figure 2 depicts the results of fitting loss curves with power-law functions for 774M Decoder-only LMs (GPT-2 Large), directly approximating how loss changes with each domain's data quantity. **X-axis depicts the quantity of domain data relative to the original amount before perturbation where 1.0=100% (original data amount), 0.33=33% (changed to 1/3x than before), 3.0=300% (changed to 3x than before).** We have updated the caption in the manuscript.
> >
> > ---
> >
> > ## Typos, formatting, visualization issues
> >
> > Thanks for pointing out these issues in the manuscript. We have revised and fixed all the listed issues together with some others. We are appreciative of the time and effort. We believe this has improved the clarity and presentation of the manuscript.

---

> ### Author Response · Authors · 2024-11-23
> **Rebuttal period ends soon–we anticipate your feedback!**
>
> Dear Reviewer JgHF,
>
> As the rebuttal/author discussion period ends in a few days, we sincerely look forward to your feedback. The authors are deeply appreciative of your valuable time and efforts spent reviewing this paper and helping us improve it.
>
> It would be very much appreciated if you could once again help review our responses and additional results and let us know if these address or partially address your concerns and if our explanations are heading in the right direction :)
>
> Please also let us know if there are any further questions or comments about this paper. We strive to consistently improve the paper and it would be our pleasure to have your precious feedback!
>
> Kind Regards,\
> Authors of Submission1690

---

> > ### Author Response · Authors · 2024-12-01
> > **Extended rebuttal period ends in 2 days–we anticipate your feedback! 🙂**
> >
> > Dear Reviewer JgHF,
> >
> > With the extended rebuttal/author discussion period ending in 2 days, we sincerely look forward to your feedback. The authors are deeply appreciative of your valuable time and efforts spent reviewing this paper and helping us improve it.
> >
> > It would be very much appreciated if you could once again help review our responses and additional results and let us know if these address or partially address your concerns and if our explanations are heading in the right direction :)
> >
> > Please also let us know if there are any further questions or comments about this paper. We strive to consistently improve the paper and it would be our pleasure to have your precious feedback!
> >
> > Kind Regards,\
> > Authors of Submission1690

---

> > > ### Author Response · Authors · 2024-12-02
> > > **Extended rebuttal period ends on Monday (Dec. 2)–we anticipate your feedback! 🙂**
> > >
> > > Dear Reviewer JgHF,
> > >
> > > With the extended rebuttal/author discussion period ending on Monday (Dec. 2), we sincerely look forward to your feedback. The authors are deeply appreciative of your valuable time and efforts spent reviewing this paper and helping us improve it.
> > >
> > > It would be very much appreciated if you could once again help review our responses and additional results and let us know if these address or partially address your concerns and if our explanations are heading in the right direction :)
> > >
> > > Please also let us know if there are any further questions or comments about this paper. We strive to consistently improve the paper and it would be our pleasure to have your precious feedback!
> > >
> > > Kind Regards,\
> > > Authors of Submission1690

---

> > > > ### Author Response · Authors · 2024-12-03
> > > > **(Last Call) Extended rebuttal period ending **Today**–we anticipate your feedback! 🙂**
> > > >
> > > > Dear Reviewer JgHF,
> > > >
> > > > The authors are deeply appreciative of your valuable time and efforts spent reviewing this paper and helping us improve it. **With the extended rebuttal/author discussion period ending *Today*, we sincerely look forward to your final feedback.**
> > > >
> > > > It would be very much appreciated if you could once again help review our responses and additional results and let us know if these address or partially address your concerns and if our explanations are heading in the right direction :)
> > > >
> > > > Please also let us know if there are any further questions or comments about this paper. We strive to consistently improve the paper and it would be our pleasure to have your precious feedback!
> > > >
> > > > Kind Regards,\
> > > > Authors of Submission1690

---

### Official Review · Reviewer_mxrW · 2024-11-05

**Soundness:** 4
**Presentation:** 3
**Contribution:** 4
**Rating:** 8
**Confidence:** 4

**Summary:**

This paper studies the problem of predicting optimal data mix for a given compute budget (i.e., fixed total token count and model size). A key challenge here is that the optimal domain weighting may change at different scale, hence it is inaccurate to use smaller models to predict large model performance, while solving the optimization problem at the large model scale directly is computationally infeasible (requires multiple retraining).

The paper proposes a method that work on one domain at a time by fixing the rest of the data constant (hence the loss is constant for other domains too), and estimated a scaling law per domain. The power law parameters $\gamma_i$ and $l_i$ can be easily estimated, which approximate a regular data scaling function where $l_i$ is the irreducible loss of that domain.

After the power law of each domain is found, the final objective is to mix the data so that the loss is minimize while keeping sum of the tokens reaches the budget, which becomes a convex function that can be solved efficiently. This gives the DDO method. The different $\gamma_i$ explains why there is a differet mix at different stage.

A method "AutoScale" is further proposed to obtain the data weight of a larger token budget, by iteratively mxing two data weights at different scale to create the weights of the next one.

The proposed approach is tested on models like GPT-2 (autoregressive) and BERT (bidirectional), showing improved convergence rates and downstream task performance. Empricially, the results show AutoScale’s ability to shift data weights, favoring diverse data sources like CommonCrawl at larger scales while reducing reliance on traditionally high-quality, standard-format data such as Wikipedia. These findings match the empricial findings of the data weights used for prior succesful models such as Llama.

**Strengths:**

- This paper analyzes an important problem, data weighting of LLM training, which can improve the training efficiency with reasonable cost. It also presents an actionable algorithm for LLM training.
- The proposed method assumes a power law formulation which makes the data weighting problem practically solvable. It is important to point out that data weights is scale dependent.
- The empirical results and findings on the corpus weighting align with common belief of the community, such that further up-weighting high quality source is less effective, and books and web documents continue to be important at larger scale. This shows that the proposed method has strong explanatory ability.
- The experiment is quite thorough, considering the cost for training models is quite high even at small scales.

**Weaknesses:**

- I wonder if carbon footprint of the experiments here should also be reported
- The presentation is good but can still be improved. The core method part can be improved with additional intuitive explanations and better use of notations. Further, I have noted down some minor typo/notation errors:

Typo or notation:
  - L258: $N_i^+2$ should probably be $N_i^+$
  - L526: Ecoder -> Encoder
  - L200: probably should better use $N_i$ instead of $|S_i|$
  - L281: $w_i'*N'$ appears twice.
  - Figure 2 caption: meaning of (original = 1) is a bit unclear
  - L380: $N$ is missing in the first equation.

**Questions:**

- I am a bit unclear about your definition of "equivalent data size" at L243, what's the equivalence about (i.e., which size and which size)? Note that I understand the meaning of $N_I^0$, just wondering the terminology here.
- Maybe I missed something, but how do one control the budget for the next $N^(3)$? It seems the amount of tokens is defined by the initial weights of  $N^(1)$ and $N^(2)$. Or in other words, say I need to find a optimal weight for a total token of 300B, how should I start with $N^(1)$ and $N^(2)$?
- Adding to the prior question, if the optimal ratio of each domain follows a exponential function, after taking a few data points using AutoScale, can we simply fit the exponential function instead of using the AutoScale iterative method? You seem to be using that in Figure 1 (d). If y es, this simply answer my question above.
- While the problem of different data scale is resolved with a scaling law solution, can we also use a similar approach on model scale? Even though the cost of using a small amount of data for a larger model should be within a low percentage of the total training cost, setting up the experiment for the larger scale is non-trivial. It'd be nice to have a function that can predict the loss across model scales.

---

### Official Review · Reviewer_Qo6P · 2024-11-08

**Soundness:** 3
**Presentation:** 3
**Contribution:** 3
**Rating:** 6
**Confidence:** 4

**Summary:**

The authors address an interesting topic in this paper: a method for automatically optimizing the mixture proportions of pretraining data domains when training language models.
They begin by formulating the optimal mixing problem as a bi-level optimization and then propose the Direct Data Optimization (DDO) algorithm to formalize the relationship between optimal data compositions and training data scales. Using DDO, they conduct empirical studies to optimize domain weights at various training data scales, demonstrating that the optimal data composition varies with the scale of the training data. Finally, they introduce AutoScale, which automatically predicts optimal training data compositions at larger scales based on compositions optimized at smaller scales.
Additionally, their evaluation of AutoScale on both decoder-only and encoder-only models demonstrates its ability to achieve computational savings.

**Strengths:**

1. AutoScale presents an interesting idea that distinguishes it from previous work, demonstrating that the optimal weights are only effective at the scale they were optimized for and become suboptimal when applied to other scales. It offers a practical method for automatically and efficiently determining domain weights when train large language models.
2. The experiments are conducted on both encoder-only and decoder-only models and shows good results on decoder-only model.
3. The work is supported by both empirical experiments and mathematical formulations. Additionally, the diagram in the paper is well-designed and effectively conveys the underlying concepts.

**Weaknesses:**

1. The experimental setup is not entirely convincing:

* The models used (a 774M decoder-only model and a 110M encoder-only model) are relatively small compared to today’s large language models, making it difficult to gauge performance at a larger scale.
* The data size is limited to 3B, 5B, and 10B tokens, with results in Table 1 only reflecting the 3B set.
* Figure 3(b) lacks explanation, and the cola baseline and DDO performance seems unusually low, falling below random guessing (0.5). Also, stsb baseline seems low too.

2. The evaluation of downstream tasks could be expanded. It would be helpful to see the models' performance on more complex tasks, such as mathematical problem-solving.

**Questions:**

1. If I understand correctly, for the downstream tasks, the evaluation metric used is perplexity. Why is perplexity chosen as the metric instead of one that is specific to the dataset or task itself?
2. Is there any potential explanation for why AutoScale doesn't perform as well on encoder-only models compared to decoder-only models?

---

> ### Author Response · Authors · 2024-11-23
> **Rebuttal period ends soon–we anticipate your feedback!**
>
> Dear Reviewer Qo6P,
>
> As the rebuttal/author discussion period ends in a few days, we sincerely look forward to your feedback. The authors are deeply appreciative of your valuable time and efforts spent reviewing this paper and helping us improve it.
>
> It would be very much appreciated if you could once again help review our responses and additional results and let us know if these address or partially address your concerns and if our explanations are heading in the right direction :)
>
> Please also let us know if there are any further questions or comments about this paper. We strive to consistently improve the paper and it would be our pleasure to have your precious feedback!
>
> Kind Regards,\
> Authors of Submission1690

---

> > ### Comment · Reviewer_Qo6P · 2024-11-29
> >
> > Thank you for the rebuttal. I agree that this paper's idea is interesting and could serve as valuable inspiration for further research in this area. While the experiments are relatively small in scale, the scope of the work is already comparable to some previous studies. Based on this, I would like to increase my score for the contribution of this work.
> >
> > However, I still have some concerns:
> >
> > 1. Regarding the task performance for DDO results in Figure 3(b): For CoLA, the original evaluation metric is Matthews Correlation Coefficient (MCC), but many research papers report results as accuracy. For stsb, even when using Pearson Correlation Coefficient (PCC), the baseline appears extremely low to me. Could you clarify the reason for this significant difference between uniform and DDO weights for stsb task, given that the differences for other GLUE tasks are not as pronounced?
> >
> > 2. Another concern I have is with Takeaway 3: “AUTOSCALE-predicted weights consistently outperform any baseline with a
> > 28% to 38% margin and demonstrate advantageous performance on downstream tasks.” This claim seems overstated and potentially confusing. It gives the impression that there is a 28%–38% performance improvement, whereas it actually refers to the speedup for decreasing validation perplexity. I think the phrasing in the abstract is more accurate and better reflects the findings.

---

> ### Author Response · Authors · 2024-11-30
> **2nd-round responses by authors**
>
> The authors sincerely appreciate the reviewer's consideration and value the reviewer's feedback! We strive to consistently improve the manuscript and serve its goal to contribute to the research community.
>
> ---
> ### Concern 1(a): accuracy vs. MCC
>
> **Re:** We fully understand the reviewer's question. In the current literature, **there appears no consensus on the primary evaluation metrics for GLUE tasks.** The original paper proposing the GLUE benchmark, *[GLUE: A Multi-Task Benchmark and Analysis Platform for Natural Language Understanding]*, lists Matthews corr. as the evaluation metric for "cola", Pearson/Spearman corr. for "stsb", acc./F1 and "mrpc" and "qqp", and accuracy for other tasks. **This is how we chose evaluation metrics in this work.**
>
> Nonetheless, **we did notice the considerable wide range of choices for evaluation metrics of the GLUE benchmark among published works**, including seminal papers. For example, in Google's original paper proposing the BERT model, *[BERT: Pre-training of Deep Bidirectional Transformers for Language Understanding]*, the evaluation on GLUE benchmark excludes the "wnli" task and reports F1 scores for "qqp" and "mrpc", Spearman correlations for "stsb", and accuracy scores for the other tasks. *[Data Selection for Language Models via Importance Resampling]* reports accuracies for all tasks. **We are neutral about which metric to use and agree that unifying the experiment setup will be beneficial**, which would cause less confusion and allow better comparison between different works.
>
> ---
>
> ### Concern 1(b): low baseline performance for "stsb"
>
> **Re:** While developing this work, we noticed the performance on task "stsb" demonstrates the phenomenon often referred to as **"emergent ability",  which describes the scenarios where the task performance of an LLM improves abruptly in an unpredictable manner while increasing the model size/compute budget** (ref: *[Emergent Abilities of Large Language Models]*). In this work, we observed the performance on task "stsb" increases sharply when the training budget reaches certain thresholds. In the experiment, **for the same training data budget**, DDO-optimized domain weights led to higher training efficiency and the trained model observed a sharp improvement in "stsb" performance. On the contrary, **the model trained with uniform domain weights did not reach the threshold for performance hike**, making the performance discrepancy appear wide.
>
> It was not the intention of this work to design experiments in this specific manner. Rather, as we stressed in the review responses, we note that task performance is generally less predictable and **better used as a qualitative measure.** Task performance does not scale smoothly with computing budget/model size and it is currently an open question whether/how task performance for LLMs can be predicted. Perplexity/loss, on the contrary, scales stably and serves as a better indicator to track the model's training progress/capability. **We recommend considering validation perplexity as the primary metric for efficiency research on LLM pre-training, which allows more precise quantitative comparisons.**
>
> ---
>
> ### Concern 2: relative improvements in Takeaway 3
>
> **Re:** We appreciate the feedback. **We will revise the manuscript to clearly state that the improvements are in the training efficiency where AutoScale-predicted weights train the model 28%–38% *faster* than baselines.**

---

> > ### Author Response · Authors · 2024-12-03
> > **(Last Call) Extended rebuttal period ending *Today*–we anticipate your feedback! 🙂**
> >
> > Dear Reviewer Qo6P,
> >
> > The authors are deeply appreciative of your valuable time and efforts spent reviewing this paper and helping us improve it. **With the extended rebuttal/author discussion period ending *Today*, we sincerely look forward to your final feedback.**
> >
> > It would be very much appreciated if you could once again help review our responses and let us know if these address or partially address your concerns and if our explanations are heading in the right direction :)
> >
> > Please also let us know if there are any further questions or comments about this paper. We strive to consistently improve the paper and it would be our pleasure to have your precious feedback!
> >
> > Kind Regards,\
> > Authors of Submission1690

---

### Meta-Review · Area_Chair_C33m · 2024-12-20

**Metareview:**

This paper attempts to tackle the challenging problem of optimizing data mixing ratios for language model training across different scales. The authors propose a computationally efficient approach by first analyzing domains independently to derive power law scaling parameters, then solving a convex optimization problem to determine optimal mixing ratios. They further introduce AutoScale, an iterative method for predicting mixing ratios at larger scales by interpolating between known solutions, addressing the practical challenge that optimal data mixing strategies often evolve with model scale.

The paper's key strengths lie in its decomposition of an intractable optimization problem into manageable components and its principled mathematical framework. The empirical validation across both autoregressive (GPT-2) and bidirectional (BERT) architectures demonstrates practical utility. The work effectively transforms what has historically been a largely empirical process into a more systematic approach, potentially reducing the computational overhead in developing large language models.

This paper exhibits several notable limitations that could be improved. First, as mentioned in most reviewers, the experimental scope is constrained by using only 3B tokens (in Table 1, the most important table to show the improvements), which is insufficient for a paper studying language model pre-training and should be extended to larger token counts. In practice, we usually use Training-Compute Optimal data allocation for a model pre-training, and it should be nearly 10B tokens for a 774M GPT model. Second, the discussion with previous work is inadequate, particularly in its omission of the Data Mixing Laws (DML) work, which pioneered the application of scaling laws to data mixing - this relationship should be explicitly addressed to properly contextualize the current contributions. Additionally, the methodology used to establish scaling laws raises concerns, as traditional scaling law studies typically analyze thousands of data points to demonstrate consistent patterns, while this study's limited data points (at most 1.2B token budget) make it difficult to confidently characterize the observed relationships as scaling laws. Finally, I suggest reconsidering the term "optimal data composition" used throughout the paper, as it may be over claiming the findings. True optimality would require exhaustive testing of all possible data compositions, which is nearly impossible. A more precise term, such as "effective data composition" or "improved data composition" would better reflect the actual extent of the optimization conducted. Additionally, the absence of comparisons against DoReMi on the original Pile dataset leaves some questions on the experiments, also.

Overall, reviewers have a mixed feeling of this paper. I have carefully read all the reviews and the author rebuttal, and have read the paper again. While the mathematical framework and formulation are commendable, the experimental validation requires substantial strengthening.

**Additional Comments On Reviewer Discussion:**

While reviewing this paper, it's worth noting that one reviewer (JgHF) did not participate in the rebuttal discussion, and consequently, their opinion carried less weight in the final decision-making process. The remaining reviewers engaged with the authors' rebuttal and contributed to a constructive discussion. After careful examination of both the paper and the arguments raised by reviewer Ad96, the area chair acknowledges the paper's limitations in its experimental and makes the final decision.

---

### Decision · Program_Chairs · 2025-01-22

Reject